# A Theory-Driven Approach to Inner Product Matrix Estimation for Incomplete Data: An Eigenvalue Perspective

## Abstract

Addressing the critical challenge of data incompleteness in inner product matrix estimation, we introduce a novel eigenvalue correction method designed to precisely reconstruct true inner product matrices from incomplete data. Utilizing random matrix theory, our method adjusts the eigenvalue distribution of the estimated inner product matrix to align with the ground-truth. This approach significantly reduces estimation errors for both inner product matrices and the derived Euclidean distance matrices, thereby enhancing the effectiveness of similarity searches on incomplete data. Our method surpasses traditional data imputation and similarity calibration techniques in both maximum inner product search and nearest neighbor search tasks, demonstrating marked advancements in managing incomplete data. It exhibits robust performance across various missing rates and diverse scenarios.

## CCS Concepts

• **Computing methodologies** → **Machine learning**; • **Mathematics of computing** → *Distribution functions*; • **Information systems** → *Information retrieval*.

## Keywords

Inner Product Matrix Estimation, Incomplete Data, Eigenvalue Distribution, Random Matrix Theory

### ACM Reference Format:

Anonymous Author(s). 2025. A Theory-Driven Approach to Inner Product Matrix Estimation for Incomplete Data: An Eigenvalue Perspective.

## 1 Introduction

In information retrieval, accurately calculating inner product between data samples is crucial [27]. When data is fully observed, this calculation is straightforward. However, incomplete data, which frequently occurs during collection and transformation, prevents direct computation of pairwise inner products. As a result, estimation becomes necessary, often leading to a significant decrease in the accuracy of inner product measurements [8, 18, 24]. This challenge is amplified when a large portion of the data is missing, making it both more difficult and more critical to obtain a high-quality inner product matrix for downstream applications. To address this issue, we propose a simple, effective, and robust approach for improving the accuracy of inner product estimation on incomplete data using

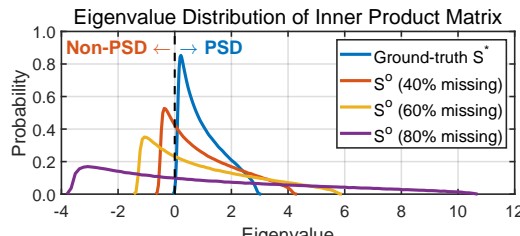

**Figure 1: A motivating example: Eigenvalue distribution divergence of inner product matrix $S^o$ on incomplete data $X^o$.**

eigenvalue analysis, benefiting applications such as maximum inner product search (MIPS) and nearest neighbor search (NNS).

Traditional methods for handling incomplete data, such as **data imputation** [9, 17], rely heavily on assumptions about the underlying data structure. For instance, many matrix completion methods assume the data follows a low-rank [4, 10] or high-rank [6] structure, while optimal transport-based methods [21, 28] assume that the distribution between observed and missing data is aligned. However, these approaches have two main drawbacks: **(1) Inaccurate Estimation:** Their primary goal is to recover the missing data, not to ensure accurate inner product calculations, which often leads to errors in the resulting inner product matrices [18]; and **(2) Performance Degradation:** Their effectiveness, particularly in inner product estimation and retrieval accuracy, declines significantly as the proportion of missing data increases, rendering them ineffective in high-missingness scenarios [26].

Alternatively, a series of optimization approaches, known as **similarity calibration** [16, 18, 26], emphasize the importance of ensuring that the inner product matrix is positive semi-definite (PSD) [22]. Given incomplete data $X^o = [x_i^o]_{i=1}^n \in \mathbb{R}^{d \times n}$ with $n$ samples, these techniques bypass data imputation by starting with an initial inner product matrix $S^o \in \mathbb{R}^{n \times n}$, where $S_{ij}^o$ denotes the estimated inner product between $x_i^o$ and $x_j^o$. The matrix $S^o$ is then calibrated to the nearest PSD matrix by solving the optimization problem, i.e., $\min_{S \succeq 0} \|S - S^o\|_F^2$. Due to their reliance on the PSD property, they face two major limitations: **(1) Limited Applicability:** If $S^o$ is already PSD, no further improvement can be made; and **(2) Limited Improvement:** Simply restoring the PSD property often fails to capture the underlying structure of the true inner product matrix $S^*$, leading to only marginal improvements. The core issue is that these methods do not directly address the discrepancy between the initial estimate $S^o$ and the true matrix $S^*$, specifically $\|S^o - S^*\|_F$, which motivates us to design a more reliable approach.

Our goal goes beyond merely ensuring the PSD property; we aim to accurately reconstruct $S^*$. Achieving this requires consistent estimation of both eigenvalues and eigenvectors, yet accurately estimating high dimensional eigenvectors is notably challenging [2, 14]. Consequently, our focus narrows to the estimation of eigenvalues.

In Fig. 1, we examine the eigenvalue distribution and uncover a key insight: ***as the missing rate increases, the eigenvalues of $S^o$ increasingly diverge from those of the ground truth matrix $S^*$.*** This underscores the necessity for a method that adjusts $S^o$'s eigenvalues to more closely match the empirical spectral distribution (ESD) with that of the ground truth $S^*$.

To accurately recover eigenvalues, it is therefore natural to ask the following questions regarding the variations in eigenvalues:

> **Q1.** *How does missingness alter eigenvalues from $S^*$ to $S^o$?*
> **Q2.** *How to correct $S^o$'s eigenvalues to recover those of $S^*$?*

This paper explores these bidirectional questions within the context of the inner product matrix. For Q1, we theoretically analyze the impact of missing data on the eigenvalue distribution for both i.i.d. and non-i.i.d. data, providing a clear explanation for the observed eigenvalue distribution divergence, using the Marchenko-Pastur (MP) Law from random matrix theory. In response to Q2, we propose a series of algorithms designed to accurately correct eigenvalues for i.i.d., non-i.i.d., and real-world data, backed by solid theoretical support.

Our contributions are summarized as follows:

• **Theory-Driven Approach:** Moving beyond merely ensuring the PSD property [16, 18, 26], we introduce a fundamentally different approach that accurately corrects the eigenvalue distribution of inner product matrices. Leveraging the MP Law, we propose an optimal eigenvalue correction strategy for incomplete i.i.d. data in Section 3, supported by theoretical bounds in Theorems 4 and 7. This strategy is extended into a practical algorithm for non-i.i.d. data in Section 4, requiring no assumptions about missing mechanisms and effectively aligning $S^o$'s eigenvalues with those of $S^*$.

• **Robust Performance:** We present simple yet effective algorithms that provide high-quality estimations of inner product matrices, even under a wide range of missing rates. Extensive experimental results demonstrate the robust performance of our eigenvalue correction approaches across several key areas: **(1) accurate estimation** of both inner product and Euclidean distance matrices, **(2) stable performance** in downstream applications, i.e., maximum inner product search and nearest neighbor search tasks, even with high missing rates, and **(3) broad applicability** across various data types and missingness scenarios, consistently outperforming traditional data imputation and similarity calibration methods.

**Notations.** Complete matrices (vectors) are denoted by $X$ ($x$) and observed matrices (vectors) are denoted by $X^o$ ($x^o$), which may contain missing values. If no missing values, $X^o = X$ and $x^o = x$. $S$ denotes the normalized inner product matrix and $D$ denotes the squared Euclidean distance matrix.

## 2 Preliminaries

### 2.1 Intuitive Estimation of Inner Product

Estimating pairwise inner products is challenging with incomplete data. [18, 26] provided an intuitive estimate for inner products on partially observed data, denoted as $x^o, y^o \in \mathbb{R}^d$. As depicted in Fig. 2, employing a non-empty index set $I \subseteq \{1, \cdots, d\}$ that identifies jointly observed features, the normalized inner product

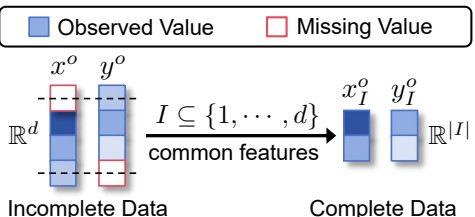

**Figure 2: Intuitive estimation of pairwise inner product.**

can be estimated unbiasedly within the $|I|$-dimensional space by:

$$\frac{1}{d}x^{o\top}y^o \approx \frac{1}{|I|}x_I^{o\top}y_I^o =: s^o(x^o, y^o). \tag{1}$$

For observed data matrix $X^o = [x_i^o]_{i=1}^n \in \mathbb{R}^{d \times n}$ with $n$ samples, the normalized inner product matrix is intuitively estimated by $S^o = [s^o(x_i^o, x_j^o)]_{i,j=1}^n \in \mathbb{R}^{n \times n}$.

### 2.2 Similarity Calibration Method's Limitations

The most closely related work in similarity estimation is the similarity calibration method [16, 18, 26], which aims to find the nearest PSD matrix for the initial estimate $S^o$ by solving $\hat{S} := \arg\min_{S \succeq 0} \|S - S^o\|_F^2$. However, its reliance on the PSD property leads to the following inherent limitations:

**(1) Limited Applicability:** Its effectiveness is contingent on $S^o$ being non-PSD. As illustrated in Fig. 3(a), with a small missing rate (e.g., 20%) and a large $\frac{d}{n}$ (e.g., 10), $S^o$ is likely to be PSD and $\hat{S} = S^o$. The non-PSD requirement precludes further improvement, thereby narrowing the method's applicability.

**(2) Limited Improvement:** As depicted in Fig. 3(b), this technique derives $\hat{S}$ by setting all negative eigenvalues of $S^o$ to zero. The resulting eigenvalue distribution (yellow line) remains significantly distant from the ground truth (blue line), inadequately capturing the true distribution and yielding marginal improvements.

**(3) Limited Distance Estimation:** The Euclidean distance matrices derived from the calibrated inner product matrices often show large estimation errors, resulting in weak performance in nearest neighbor search tasks. This is likely due to PSD optimization altering the intrinsic structure of the inner product matrices, which degrades the quality of the derived distance matrices.

These limitations motivated us to develop a new method that accurately recovers the inner product matrix, especially for the eigenvalue distribution, without relying on the PSD property.

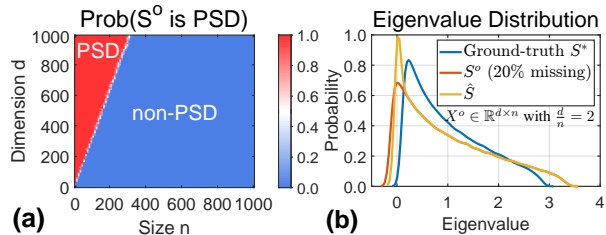

**Figure 3: Limitations of similarity calibration method: (a) Limited Applicability and (b) Limited Improvement. We consider $X^o \in \mathbb{R}^{d \times n}$, where $x_{ij}^o \overset{i.i.d.}{\sim} \mathcal{N}(0, 1)$ with 20% missing.**

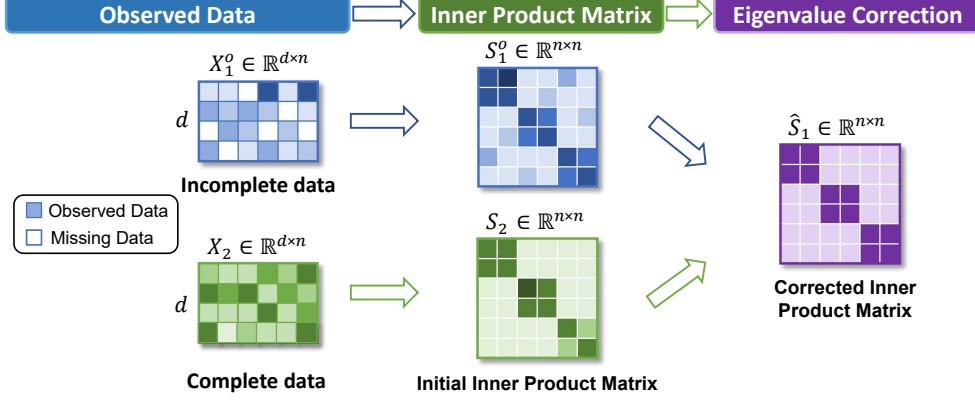

**Figure 4: A diagram of our proposed eigenvalue correction approach.**

## 3 Inner Product Estimation for I.I.D. Data

We aim to accurately reconstruct the true inner product matrix $S^*$, starting with the intuitive estimate $S^o$. The key question, from the perspective of eigenvalues, is understanding the relationship between the eigenvalues of $S^o$ and those of $S^*$. To explore this, we focus on simple cases of i.i.d. data, where tools from random matrix theory can be effectively used to study eigenvalue distributions.

In this section, we first explore the true eigenvalue distribution for complete i.i.d. data (Section 3.1). Next, we provide a theoretical analysis of the eigenvalue distribution for incomplete i.i.d. data (Section 3.2), offering explanations for the phenomenon of eigenvalue distribution divergence. Finally, we propose a novel eigenvalue correction approach for incomplete i.i.d. data (Section 3.3) that accurately recovers the true eigenvalues, supported by a rigorous optimality analysis (Section 3.4).

### 3.1 Inner Product of Complete I.I.D. Data

Consider fully observed i.i.d. data $X = [x_{ij}]_{i,j=1}^{n} \in \mathbb{R}^{d \times n}$ with zero mean and finite variance. The true normalized inner product matrix is defined as $S^* = X^\top X/d \in \mathbb{R}^{n \times n}$, whose eigenvalue distribution can be well described by the Marchenko-Pastur (MP) Law [19] from random matrix theory. Based on the MP Law, the convergence of the empirical spectral distribution (**ESD**) $F_n^*(x) \equiv F^*(x) = \frac{1}{n} \sum_{1 \leq i \leq n} \mathbf{1}\{\lambda_i^* \leq x\}$ of $S^*$, is established in Lemma 1. Here, $\lambda_i^*$ represents the $i$-th eigenvalue of $S^*$, ordered as $\lambda_1^* \geq \cdots \geq \lambda_n^*$.

**Lemma 1 (Eigenvalue Distribution for Complete I.I.D. Data [19]).** *Consider $X = [x_1, \ldots, x_n] \in \mathbb{R}^{d \times n}$, where the entries are i.i.d. random variables with mean 0 and variance $\sigma^2 < \infty$. As $d, n \to \infty$ with $d/n \to c > 0$, the empirical spectral distribution (**ESD**) $F^*$ of $S^*$ almost surely converges weakly to the limiting spectral distribution (**LSD**) $\mu^*$. The LSD $\mu^*$ is supported on the interval:*

$$[\lambda_-^*, \lambda_+^*] = [\sigma^2(1 - c^{-1/2})^2, \sigma^2(1 + c^{-1/2})^2],$$

*with the density function:*

$$f^*(x) = \frac{c}{2\pi\sigma^2} \frac{\sqrt{(\lambda_+^* - x)(x - \lambda_-^*)}}{x} \mathbf{1}_{x \in [\lambda_-^*, \lambda_+^*]}.$$

This lemma shows that (1) almost all non-zero eigenvalues of $S^*$ lie within the spectral support $[\lambda_-^*, \lambda_+^*]$, (2) the LSD and spectral support for complete i.i.d. data depend only on $c$, assuming the variance $\sigma^2$ is fixed, and (3) the eigenvalues of any equal-size i.i.d. data matrices $X_1, X_2, \cdots \in \mathbb{R}^{d \times n}$ drawn from the same distribution, converge to the same limiting spectral distribution, which motivates us to design algorithms for more general data in Section 4.

### 3.2 Inner Product of Incomplete I.I.D. Data

Considering partially observed i.i.d. data $X^o = [x_{ij}^o]_{i,j=1}^{n} \in \mathbb{R}^{d \times n}$, we simplify our theoretical analysis by focusing on a missing completely at random (**MCAR**) scenario, where each entry $x_{ij}^o$ is uniformly missing with probability $r \in (0, 1)$, representing the missing rate. The initial inner product matrix $S^o$ is estimated using Eq. (1).

In Theorem 2, we expand the Marchenko-Pastur Law to theoretically determine the LSD $\mu^o$ of $S^o$, illustrating that the eigenvalue distribution of $S^o$ hinges on both $c$ and $r$, proven in Appendix A.1.

**Theorem 2 (Eigenvalue Distribution for Incomplete I.I.D. Data).** *Consider $X^o = [x_1^o, \ldots, x_n^o] \in \mathbb{R}^{d \times n}$, where the true values of $\{x_i^o\}$ are i.i.d. random variables with mean 0 and variance $\sigma^2 < \infty$, missing completely at random (MCAR) with a missing rate of $r \in (0, 1)$. As $d, n \to \infty$ with $d/n \to c \in (0, +\infty)$, the limiting spectral distribution $\mu^o$ of the initial estimate $S^o$ is supported on*

$$[\lambda_-^o, \lambda_+^o] = \left[ \frac{\sigma^2(1 - c^{-1/2})^2 - r}{1 - r}, \frac{\sigma^2(1 + c^{-1/2})^2 - r}{1 - r} \right]$$

*with the density function*

$$f^o(x) = \frac{c(1-r)^2}{2\pi\sigma^2} \frac{\sqrt{(\lambda_+^o - x)(x - \lambda_-^o)}}{(1-r)x + r} \mathbf{1}_{x \in [\lambda_-^o, \lambda_+^o]}.$$

To further explore how $r$ and $c$ influence eigenvalue distributions, we graphically depict the spectral support of $S^o$ in Fig. 5 with two key observations:

- **Impact of $r$:** The spectral support $[\lambda_-^o, \lambda_+^o]$ of $S^o$ gradually widens as the missing rate $r$ increases. This confirms the eigenvalue distribution divergence, as shown in Fig. 1, which is caused by the missing values and can be theoretically explained by Theorem 2.
- **Impact of $c$:** The upper boundary $\lambda_+^o$ increases monotonically as $c$ decreases, indicating that eigenvalue distributions become more sensitive to missing data at smaller values of $c$.

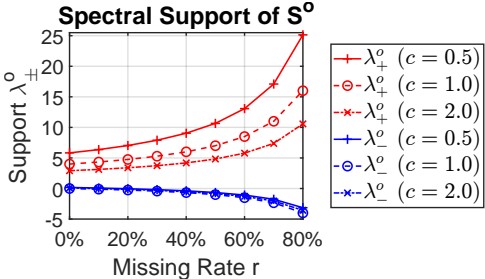

**Figure 5: The impact of $r$ and $c$ on the spectral support of $S^o$.**

Beyond the spectral support, we explore the alignment between the ESD and LSD, as shown in Fig. 6, leading to two key insights:

• **Distribution Alignment:** The ESDs of both $S^*$ and $S^o$ closely align their corresponding LSDs across varying missing rates, suggesting that the density function of the LSD can be effectively used to estimate the ESD with high accuracy.

• **Distribution Shape:** The eigenvalue distributions of $S^o$ consistently maintain a shape similar to the ground-truth distribution of $S^*$ under various missing rates, indicating a linear transformation between the LSD of $S^o$ and that of $S^*$.

This consistency of distribution shape motivates us to design a precise eigenvalue correction strategy, as detailed in Section 3.3.

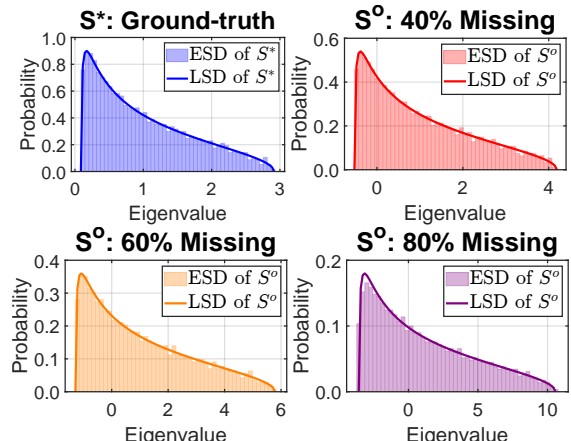

**Figure 6: Distribution alignment and consistent shape. Consider $X^o = [x_{ij}^o] \in \mathbb{R}^{2000 \times 1000}$ with $x_{ij}^o \overset{i.i.d.}{\sim} \mathcal{N}(0,1)$ and $c = 2$.**

## 3.3 Proposed Eigenvalue Correction Strategy for Incomplete I.I.D. Data

Motivated by the consistent shape in eigenvalue distributions observed in Fig. 6, we propose a novel eigenvalue correction approach to recover the true eigenvalue distribution of $S^*$ through a linear transformation. Our method corrects the eigenvalues $\{\lambda_i^o\}$ within the spectral support of $S^o$ using the following linear transformation:

$$\hat{\lambda}_i := \frac{\lambda_+^* - \lambda_-^*}{\lambda_+^o - \lambda_-^o} \cdot (\lambda_i^o - \lambda_-^o) + \lambda_-^* = (1-r)\lambda_i^o + r. \quad (2)$$

This procedure effectively aligns the spectral support of $S^o$ with that of $S^*$, as evidenced by $\lambda_\pm^* = (1-r)\lambda_\pm^o + r$. However, it is crucial to note that in scenarios where $d < n$, $S^*$ contains $(n-d)$

zero eigenvalues outside its support. Therefore, in cases where $d < n$, we adjust the smallest $(n-d)$ eigenvalues of $S^o$ to zero. The complete Algorithm 1 is summarized as follows.

---

**Algorithm 1 Eigenvalue Correction for I.I.D. Data**

---

**Input:** $X^o \in \mathbb{R}^{d \times n}$: an incomplete i.i.d. data matrix with mean 0 and variance $\sigma^2 < \infty$; $r$: the missing rate of MCAR.
**Output:** $\hat{S} \in \mathbb{R}^{n \times n}$: the corrected inner product matrix.
1: Calculate the initial estimate $S^o$ via Eq. (1).
2: Perform eigen-decomposition $S^o = U\Lambda U^\top$ with $\Lambda = \text{Diag}(\lambda_1^o, \cdots, \lambda_n^o)$ and $\lambda_1^o \geq \cdots \geq \lambda_n^o$.
3: **if** $d < n$ **then**
4:    $\hat{\lambda}_i \leftarrow (1-r)\lambda_i^o + r$ for $1 \leq i \leq d$;
5:    $\hat{\lambda}_i \leftarrow 0$ for $d+1 \leq i \leq n$.
6: **else if** $d \geq n$ **then**
7:    $\hat{\lambda}_i \leftarrow (1-r)\lambda_i^o + r$ for $1 \leq i \leq n$.
8: **end**
9: Compute the eigenvalue matrix $\hat{\Lambda} = \text{Diag}(\hat{\lambda}_1, \cdots, \hat{\lambda}_n)$.
10: **Return** $\hat{S} = U\hat{\Lambda}U^\top$.

---

## 3.4 Optimality Analysis

On the recovery of true eigenvalue distribution, we prove the optimality of the proposed correction strategy in Theorem 3. Theoretically, the consistent distribution patterns of $S^o$ and $S^*$ originate from the linear transformation relationship between their probability density functions (PDFs) $f^*(x)$ and $f^o(x)$, as defined in Lemma 1 and Theorem 2, respectively. The proof is provided in the Appendix A.2.

**Theorem 3 (Optimality of Eigenvalue Correction Strategy).** *Given incomplete i.i.d. data $X^o$ with MCAR, the linear transformation $\lambda_i^o \mapsto \hat{\lambda}_i := (1-r)\lambda_i^o + r$ is the optimal transformation to reconstruct the spectral distribution of $S^*$, in the sense that almost surely $|\hat{F}(x) - F^*(x)| \to 0$ for any $x \in \mathbb{R}$, where $\hat{F}(x)$ and $F^*(x)$ are distribution functions corresponding to $\{\hat{\lambda}_i\}$ and $\{\lambda_i^*\}$, respectively.*

Theorem 3 illustrates our capability to precisely align all eigenvalues $\{\lambda_i^o\}$ with $\{\lambda_i^*\}$ for any non-zero missing rate $r$. This marks a significant advancement over similarity calibration methods [16, 18, 26], which only partially correct negative eigenvalues and rely on a non-PSD $S^o$ under a large missingness.

Regarding the quality of inner product estimation, while previous works [16, 18, 26] assert that $\|\hat{S} - S^*\|_F \leq \|S^o - S^*\|_F$, our approach achieves a significantly tighter error bound in Theorem 4 (proven in the Appendix A.3), indicating a more substantial improvement.

**Theorem 4 (Error Bound of Inner Product Estimation).** *Given incomplete i.i.d. data $X^o$ with MCAR, for any small constant $\varepsilon$, it holds with probability $(1 - o(1))$ that $\|\hat{S} - S^*\|_F \leq (\eta_S + \varepsilon)\|S^o - S^*\|_F$, where $\eta_S = \sqrt{1 - \frac{r^2 c^{-1}}{(2+c^{-1})(1-r)^2 + 2r(1-r) + c^{-1}}} \in (0,1)$.*

**Remark.** *Achieving precise recovery of $S^*$ is generally challenging, as consistent estimation of $S^*$'s eigenvectors from $S^o$ is not feasible without additional information or a specific covariance structure [14]. Unlike previous works [16, 18, 26] that focus on restoring the PSD property, our approach aims to accurately reconstruct the true spectral distribution of $S^*$, yielding estimates with significantly reduced error.*

## 4 Inner Product Estimation for Non-I.I.D. Data

Beyond i.i.d. data, our theory can be extended to more general cases of non-i.i.d. data, where features are correlated and samples are independently and identically distributed. As is well known, the complex structure of non-i.i.d. data makes theoretical analysis more challenging, as we lack powerful tools available for non-i.i.d. data.

To effectively model non-i.i.d. data, we first investigate separable data, a generalized version of i.i.d. data that can be analyzed using random matrix theory. We then extend our theoretical insights to more general real-world data and propose a simple yet effective approach to correct eigenvalue distributions, without assuming any specific missing mechanism. This approach shows strong and robust performance in the empirical validations presented in Section 6.

### 4.1 Inner Product of Separable Data

We begin with separable data, a generalized form of i.i.d. data studied in random matrix theory. To simplify the analysis, we assume the data is missing completely at random (MCAR) and establish the relationship between the eigenvalues of the initial inner product matrix $S^o$ and the ground-truth $S^*$, as presented in Theorem 5 and proven in Appendix A.4.

**Theorem 5 (Eigenvalue Distribution for Separable Data).** *Consider non-i.i.d. separable data $X = [x_1, \ldots, x_n] \in \mathbb{R}^{d \times n}$, where $x_i = \Sigma^{1/2} z_i \in \mathbb{R}^d$, with $z_i$ having independent coordinates, $\mathbb{E}[z_i] = 0$, and $\mathrm{Cov}(z_i) = I_d$. Define $X^o$ as the incomplete version of $X$ with MCAR in a missing rate $r$, and $S^o$ as the initial inner product matrix of $X^o$. For the eigenvalues $\{\lambda_i^o\}$ of $S^o$, it holds that, for $1 \le i \le n$,*

$$\lambda_i^o - (1-r)^{-1}\lambda_i^* \xrightarrow{p} r(1-r)^{-1}\mathrm{tr}(\Sigma)/d,$$

*where $\xrightarrow{p}$ indicates convergence in probability.*

Two key insights emerge from Theorem 5. Firstly, it reveals that MCAR introduces a linear relationship: $\mathrm{Support}(S^o) \approx (1-r)^{-1}\mathrm{Support}(S^*) + r(1-r)^{-1}\mathrm{tr}(\Sigma)/d$. This relationship can be leveraged to recover the true eigenvalues and improve the inner product estimate. Secondly, it also suggests that MCAR preserves the fundamental "shape" of the LSD, modulo scaling and shifting adjustments, which is consistent with the i.i.d. case shown in Fig. 6.

### 4.2 Inner Product of Real-World Data

It is widely recognized that modeling real-world data is difficult due to their varying distributions and complex structures. Without a specific data model, it is impossible to theoretically derive the true eigenvalue distributions for incomplete real data. However, we can empirically estimate these true eigenvalue distributions using fully observed real data, which provides a foundation for further eigenvalue correction.

How can we obtain such an empirical estimate? Motivated by our theory on i.i.d. and separable data, we have both theoretically and empirically observed that ***two fully observed, equal-sized subsets $X_1$ and $X_2$ from the same dataset $\mathcal{X}$ exhibit similar eigenvalue distributions***. This implies that if $X_1^o$ is derived from $X_1$ with missing values, we can use the eigenvalue distribution of $X_2$ as an empirical estimate for that of $X_1$. In this case, the fully observed $X_2$ serves as reference data for the incomplete $X_1^o$. To formalize this observation, we present the following Theorem 6:

**Theorem 6 (Eigenvalue Distribution for Non-I.I.D. Data).** *Let $X_1, X_2 \in \mathbb{R}^{d \times n}$ be two fully-observed subsets of non-i.i.d. data from the same distribution. Assume $X_i = \Sigma^{1/2}Z_i$ ($i = 1, 2$), where $Z_i$'s elements have zero mean, share the same variance, and have finite fourth moments. Then, the empirical spectral distributions (ESDs) of $S_1 = X_1^\top X_1/d$ and $S_2 = X_2^\top X_2/d$ converge to the same limiting spectral distribution (LSD) as $d, n \to \infty$ with $d/n \to c \in (0, +\infty)$.*

### 4.3 Proposed Eigenvalue Correction Strategy for Incomplete Non-I.I.D. Data

For general real data, we also observe consistency in eigenvalue distributions, as partially supported by Theorem 6. Take the CIFAR10 image dataset [15] as an example. Consider two random subsets, $X_1$ and $X_2$, each containing 1,000 samples, and construct an incomplete $X_1^o$ from $X_1$ with 50% random missing entries. As shown in Fig. 7, we visualize the eigenvalues of their inner product matrices $S_1$, $S_2$, and $S_1^o$, and make the following observations:

• **Similar Small Eigenvalues:** $S_1$ and $S_2$ share nearly identical small eigenvalues from $\lambda_{100}$ to $\lambda_{1000}$ within the spectral support.

• **Distinct Large Eigenvalues:** $S_1$ and $S_2$ have distinct large eigenvalues from $\lambda_1$ to $\lambda_5$, which act as outliers beyond the spectral support, reflecting unique characteristics in $X_1$ and $X_2$.

• **Impact of Missingness:** $S_1^o$ and $S_1$ exhibit similar large eigenvalues from $\lambda_1$ to $\lambda_5$, but show significant differences in smaller eigenvalues from $\lambda_{100}$ to $\lambda_{1000}$, indicating that missingness has a substantial impact on the spectral support.

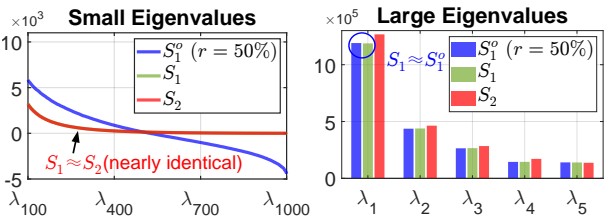

**Figure 7: Eigenvalue distributions of real data from CIFAR10.**

These insights led us to develop a universal correction strategy, presented in Algorithm 2, which does not rely on any specific missing mechanism or data distribution. To correct $S_1^o$, we use a simple yet effective approach: ***using $S_2$ as a reference, we retain $S_1^o$'s top-k largest eigenvalues and replace the rest with those from $S_2$***. This approach is inspired by the well-known spiked models [13] in random matrix theory, where signals (outliers) and the bulk (spectral support) are handled separately [14].

---

**Algorithm 2 Eigenvalue Correction for Non-I.I.D. Data**

---

**Input:** $X_1^o \in \mathbb{R}^{d \times n}$: an incomplete subset; $X_2 \in \mathbb{R}^{d \times n}$: a complete subset; $k$: top-$k$ eigenvalues (hyperparameter).

**Output:** $\hat{S}_1 \in \mathbb{R}^{n \times n}$: the corrected inner product matrix for $X_1^o$.

1: Calculate $S_1^o, S_2$ from $X_1^o, X_2$ via Eq. (1).

2: Perform $S_1^o = U_1\Lambda_1^o U_1^\top$ and $\Lambda_1^o = \mathrm{Diag}(\lambda_1^o, \cdots, \lambda_n^o)$.

3: Perform $S_2 = U_2\Lambda_2^* U_2^\top$ and $\Lambda_2^* = \mathrm{Diag}(\lambda_1^*, \cdots, \lambda_n^*)$.

4: Compute $\hat{\Lambda}_1 = \mathrm{Diag}(\underbrace{\lambda_1^o, \cdots, \lambda_k^o}_{\text{from } S_1^o}, \underbrace{\lambda_{k+1}^*, \cdots, \lambda_n^*}_{\text{from } S_2})$.

5: **Return** $\hat{S}_1 = U_1\hat{\Lambda}_1 U_1^\top$.

---

# 5 Extension

## 5.1 Extension on Scalability and Efficiency

**Scalability Analysis.** In practice, we often encounter cases where the number of incomplete samples exceeds that of complete samples. In such scenarios, we handle the unequal-sized matrices $X_1^o \in \mathbb{R}^{d \times n_1}$ and $X_2 \in \mathbb{R}^{d \times n_2}$ (with $n_1 > n_2$) by using a divide-and-conquer approach. As shown in Fig. 8, we partition $S_1^o \in \mathbb{R}^{n_1 \times n_1}$ and $S_2 \in \mathbb{R}^{n_2 \times n_2}$ into submatrices $\{S_{ij}^o\}$ and $\{S_{pq}\}$, each of size $m \times m$ (with $m \ll n_1, n_2$, assuming $n_1$ and $n_2$ are divisible by $m$). We then perform eigen-decomposition for each diagonal submatrix $S_{ii}^o$ or $S_{pp}$, and singular value decomposition for each off-diagonal submatrix $S_{ij}^o$ or $S_{pq}$, achieving **quadratic time complexity $O(mn_1^2 + mn_2^2)$** with highly parallelizable processing. Rather than correcting the entire $S_1^o$, we correct each $S_{ij}^o \in \mathbb{R}^{m \times m}$ to $\hat{S}_{ij}$ individually, and reconstruct the corrected one $\hat{S}_1 = (\hat{S}_{ij}) \in \mathbb{R}^{n_1 \times n_1}$. This approach enables more scalable processing of large datasets with a higher number of incomplete samples. The scalable **Algorithm 3** is summarized in **Appendix B.1** with a detailed example.

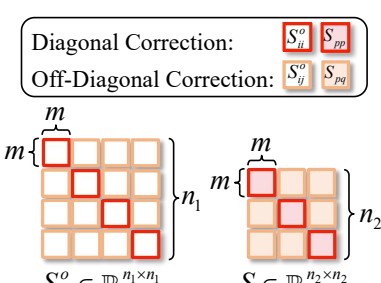

**Figure 8: Schematic diagram of the scalable Algorithm 3.**

**Efficiency Analysis.** For large datasets, the time complexity of the scalable Algorithm 3 is **quadratic**, $O(mn_1^2 + mn_2^2)$, where $m \ll n_1, n_2$. For smaller datasets, while the time complexity of Algorithms 1 and 2 is $O(n^3)$, these algorithms are highly efficient in practice. For instance, they require less than 0.1 seconds for $n = 1,000$ and less than 1 minute for $n = 10,000$. The primary computational cost arises from the eigen-decomposition of $S \in \mathbb{R}^{n \times n}$. Despite the cubic complexity, Algorithm 1 only performs eigen-decomposition once, and Algorithm 2 performs it twice, ensuring fast execution in practice. In contrast, similarity calibration methods [18, 26] require multiple eigen-decompositions in their iterative optimization procedures, leading to much longer run times.

## 5.2 Extension to Euclidean Distance

We have established a comprehensive framework for estimating the inner product on both i.i.d. and non-i.i.d data in Sections 3 and 4. It is natural to compute the Euclidean distance from the inner product. Suppose we have obtained normalized inner product $\hat{S}$ for incomplete $X^o = [x_{ij}^o] \in \mathbb{R}^{d \times n}$, the squared Euclidean distance between $x_i^o$ and $x_j^o \in \mathbb{R}^d$ can be approximated via $\|x_i^o - x_j^o\|^2 \approx d \cdot \hat{s}_{ii} + d \cdot \hat{s}_{jj} - 2d \cdot \hat{s}_{ij} =: \hat{d}_{ij}$, where $\hat{s}_{ij}$ approximates $\frac{1}{d} x_i^{o\top} x_j^o$. Then, the squared Euclidean distance matrix is obtained by

$$\hat{D} := [\hat{d}_{ij}] = \text{Diag}(d\hat{S}) \cdot J + J \cdot \text{Diag}(d\hat{S}) - 2d\hat{S}, \qquad (3)$$

where $J$ is an all-ones matrix of size $n \times n$ and $\text{Diag}(\cdot)$ is to extract a diagonal matrix. Furthermore, we derive an error bound for the Euclidean distance estimation in Theorem 7, proven in Appendix A.6.

**Theorem 7 (Error Bound of Euclidean Distance Estimation).** *Given incomplete i.i.d data $X^o$ with MCAR, there exists $\eta_D \in (0, 1)$ such that $\|\hat{D} - D^*\|_F \leq (\eta_D + \varepsilon)\|D^o - D^*\|_F$ holds with probability $(1 - o(1))$ for any small $\varepsilon > 0$, with $\eta_D$ specified in Eq. (A.13).*

**Remark.** *Our method's corrected inner product results in significantly smaller estimation errors of Euclidean distance in practice. By comparison, similarity calibration methods lack theoretical guarantees for Euclidean distance, often leading to larger errors and weaker performance in nearest neighbor search tasks.*

# 6 Experiments

To evaluate the performance, we focus on the estimation errors of both inner product and Euclidean distance matrices (Section 6.2), with applications in maximum inner product search and nearest neighbor search (Section 6.3). This is followed by a robustness analysis (Section 6.4) and application extensions (Section 6.5).

## 6.1 Experimental Setting

**Dataset.** We evaluate the performance on four benchmark datasets, covering a reasonable range of applications: **CIFAR10** [15]: a color-image dataset with colorful images of $32 \times 32$ ($d = 3,072$); **LFW** [12]: a face-image dataset with resized gray images of $64 \times 64$ ($d = 4,096$); **COIL100** [23]: an object-image dataset with resized gray images of $32 \times 32$ ($d = 1,024$); **ISOLET** [5]: a speech dataset that contains recordings of different speakers ($d = 617$).

**Data Setting.** From each dataset, we randomly select two subsets, each containing $n$ samples: one serves as the incomplete $X_1^o$, and the other as the complete $X_2$. **Our experiments focus on inner product matrix estimation for the incomplete $X_1^o$.** We report average results for 10 random seeds on a ThinkStation equipped with an Intel i7-12700 Core and 32GB RAM.

**Missing Mechanism.** For simplicity and fairness, we apply the most commonly used Missing Completely at Random (MCAR) mechanism [17] in Sections 6.2-6.4, where each entry in $X_1^o$ is replaced by the NA value with a probability $r$, known as the *missing rate*. **Crucially, our algorithms' application to real data in Algorithms 2 and 3, operates without explicit assumptions about the missing mechanism.** It proves effective across various missing mechanisms, as shown in Section 6.5.

**Baseline Methods.** Various methods designed for handling incomplete data are considered for comparison, including: **(1) Statistical Imputation: Mean** [11], $k$-nearest neighbors (**kNN**) [1]; **(2) Matrix Completion:** Singular Value Thresholding (**SVT**) [3], Kernelized Factorization Matrix Completion (**KFMC**) [6], Polynomial Matrix Completion (**PMC**) [7]; **(3) Optimal-transport-based Imputation:** Transformed Distribution Matching (**TDM**) [28]; **(4) Deep Imputation: GAIN** [25] and **MIWAE** [20]; **(5) Similarity Calibration:** Direct Matrix Calibration (**DMC**) [16], Similarity Matrix Calibration (**SMC**) [26], and Similarity Vector Calibration (**SVC**) [18]. Implementation details and hyperparameters are provided in **Appendix C**.

**Table 1: Comparison of Relative Error (RE) in inner product and Euclidean distance estimation with $n = 1,000$ samples and $80\%$ random missing. Bold shows the best result, and underline marks the second-best. Our method achieves the lowest errors.**

| Metric: $\text{RE}(X) = \frac{\|X-X^*\|_F}{\|X^*\|_F}$ ↓ | | Relative Error of $S$ | | | | Relative Error of $D$ | | | |
|---|---|---|---|---|---|---|---|---|---|
| Baseline Type | Method | CIFAR10 | LFW | COIL100 | ISOLET | CIFAR10 | LFW | COIL100 | ISOLET |
| *Statistical* | Mean (2005) | $0.958_{\pm0.000}$ | $0.958_{\pm0.000}$ | $0.957_{\pm0.000}$ | $0.958_{\pm0.000}$ | $0.814_{\pm0.001}$ | $0.811_{\pm0.000}$ | $0.808_{\pm0.003}$ | $0.810_{\pm0.001}$ |
| *Imputation* | $k$NN (2016) | $0.947_{\pm0.002}$ | $0.944_{\pm0.003}$ | $0.939_{\pm0.003}$ | $0.944_{\pm0.003}$ | $0.811_{\pm0.001}$ | $0.806_{\pm0.001}$ | $0.802_{\pm0.003}$ | $0.803_{\pm0.001}$ |
| *Matrix* | SVT (2010) | $0.865_{\pm0.002}$ | $0.866_{\pm0.002}$ | $0.869_{\pm0.003}$ | $0.880_{\pm0.002}$ | $0.790_{\pm0.001}$ | $0.791_{\pm0.001}$ | $0.789_{\pm0.002}$ | $0.795_{\pm0.001}$ |
| *Completion* | KFMC (2019) | $0.946_{\pm0.013}$ | $0.958_{\pm0.000}$ | $0.916_{\pm0.004}$ | $0.934_{\pm0.001}$ | $0.811_{\pm0.003}$ | $0.811_{\pm0.000}$ | $0.768_{\pm0.014}$ | $0.804_{\pm0.002}$ |
| | PMC (2020) | $0.841_{\pm0.010}$ | $0.924_{\pm0.003}$ | $0.733_{\pm0.022}$ | $0.841_{\pm0.005}$ | $0.743_{\pm0.009}$ | $0.802_{\pm0.003}$ | $0.789_{\pm0.254}$ | $0.769_{\pm0.004}$ |
| *OT Imputation* | TDM (2023) | $0.957_{\pm0.001}$ | $0.956_{\pm0.001}$ | $0.956_{\pm0.001}$ | $0.957_{\pm0.001}$ | $0.789_{\pm0.003}$ | $0.784_{\pm0.003}$ | $0.785_{\pm0.003}$ | $0.780_{\pm0.004}$ |
| *Deep* | GAIN (2018) | $1.074_{\pm0.167}$ | $1.200_{\pm0.197}$ | $2.053_{\pm0.289}$ | $8.313_{\pm3.783}$ | $0.432_{\pm0.065}$ | $0.504_{\pm0.064}$ | $0.418_{\pm0.036}$ | $0.327_{\pm0.056}$ |
| *Imputation* | MIWAE (2019) | $0.625_{\pm0.014}$ | $0.577_{\pm0.013}$ | $0.918_{\pm0.002}$ | $0.824_{\pm0.058}$ | $0.281_{\pm0.003}$ | $0.228_{\pm0.005}$ | $0.466_{\pm0.102}$ | $0.398_{\pm0.023}$ |
| *Similarity* | DMC (2015) | $0.225_{\pm0.007}$ | $0.228_{\pm0.003}$ | $0.488_{\pm0.012}$ | $0.696_{\pm0.009}$ | $0.742_{\pm0.007}$ | $0.662_{\pm0.004}$ | $1.315_{\pm0.014}$ | $1.903_{\pm0.017}$ |
| *Calibration* | SMC (2023) | $\underline{0.184}_{\pm0.006}$ | $\underline{0.190}_{\pm0.003}$ | $\underline{0.375}_{\pm0.010}$ | $\underline{0.508}_{\pm0.007}$ | $0.306_{\pm0.007}$ | $0.266_{\pm0.003}$ | $0.579_{\pm0.018}$ | $0.829_{\pm0.012}$ |
| | SVC (2024) | $0.226_{\pm0.013}$ | $0.220_{\pm0.003}$ | $0.447_{\pm0.022}$ | $0.631_{\pm0.021}$ | $0.490_{\pm0.005}$ | $0.428_{\pm0.003}$ | $0.901_{\pm0.010}$ | $1.331_{\pm0.012}$ |
| *Initial Estimate* | $S^o$ & $D^o$ | $0.269_{\pm0.008}$ | $0.273_{\pm0.004}$ | $0.574_{\pm0.013}$ | $0.806_{\pm0.010}$ | $\underline{0.079}_{\pm0.001}$ | $\underline{0.072}_{\pm0.000}$ | $\underline{0.214}_{\pm0.032}$ | $\underline{0.219}_{\pm0.005}$ |
| *Our Method* | EC | $\mathbf{0.156}_{\pm0.005}$ | $\mathbf{0.160}_{\pm0.003}$ | $\mathbf{0.305}_{\pm0.008}$ | $\mathbf{0.397}_{\pm0.004}$ | $\mathbf{0.049}_{\pm0.001}$ | $\mathbf{0.046}_{\pm0.001}$ | $\mathbf{0.182}_{\pm0.038}$ | $\mathbf{0.148}_{\pm0.005}$ |
| Improvement from $S^o$, $D^o$ to Ours | | $42\%_{\pm0\%}$ | $42\%_{\pm0\%}$ | $47\%_{\pm1\%}$ | $51\%_{\pm1\%}$ | $38\%_{\pm0\%}$ | $36\%_{\pm0\%}$ | $16\%_{\pm5\%}$ | $32\%_{\pm1\%}$ |

## 6.2 Evaluation on Estimation Error

To estimate a high-quality inner product matrix for incomplete data, we aim to produce an accurate estimate for the incomplete data $X_1^o$ and derive a reliable Euclidean distance matrix from it. The evaluation metric we use is **Relative Error (RE)**, which quantifies the error in the estimated matrices. The relative error of the inner product matrix $S$ and the Euclidean distance matrix $D$ is defined as:

$$\text{RE}(X) := \frac{\|X - X^*\|_F}{\|X^*\|_F} \qquad (4)$$

where $X$ denotes the estimated matrix (either inner product $S$ or Euclidean distance $D$), and $X^*$ represents the ground truth matrix.

As illustrated in Table 1, our **Eigenvalue Correction (EC)** method demonstrates superior performance across both inner product and Euclidean distance estimation compared to baseline methods, including data imputation and similarity calibration techniques.

• **Comparison with Imputation Methods: (1) Statistical Imputation:** The $k$NN method estimates missing values by averaging those of the nearest neighbors. However, the neighbor relationship can be compromised by missing values, leading to inaccurate estimates. **(2) Matrix Completion:** Methods like SVT, KFMC, and PMC perform matrix completion based on assumptions of low-rank or high-rank structures, and their performance often deteriorates when the data does not fit these assumptions. **(3) Optimal-transport-based Imputation.** TDM utilizes optimal transport to match distributions of $X$ but does not match spectral distribution of the inner product matrix $S$. **(4) Deep Imputation:** The performance of deep learning models like GAIN and MIWAE heavily relies on the quality and size of the training data. In our case, the training data is limited to only 1,000 samples from the complete data $X_2$, and the missing rate is high (80%), both of which contribute to the suboptimal performance of these deep imputation methods. In sum, imputation methods aim to recover the original data rather than the inner product matrix $S$, which may not ensure

the quality of $S$ and $D$. Additionally, imputation methods may not be applicable at high missing rates (e.g., 80%), where their performance significantly degrades, as illustrated in Fig. 9 in Section 6.4.

• **Comparison with Calibration Methods:** Similarity calibration methods, such as DMC, SMC and SVC, improve the initial estimate $S^o$ by adjusting it to the nearest PSD matrix. While this reduces the estimation error of the inner product matrix compared to $S^o$, these methods fail to capture the true eigenvalue distribution of the inner product matrix. As a result, they can distort the structure of $S$, leading to an unreliable Euclidean distance matrix with significantly larger errors than $D^o$ derived from $S^o$.

• **Improvement from $S^o$, $D^o$ to Ours:** Our method consistently improves the initial estimate $S^o$, achieving 42%-51% reductions in relative errors. This improvement is driven by our method's ability to effectively correct the eigenvalue distribution of $S^o$ to align with the ground truth, highlighting the importance of eigenvalue distribution in inner product estimation. As a result, the Euclidean distance matrices derived from our corrected inner product matrices also outperform $D^o$, with 16%-38% error reduction.

## 6.3 Evaluation on Similarity Search

We evaluate the quality of the estimated inner product matrix $S \in \mathbb{R}^{n \times n}$ and Euclidean distance matrix $D \in \mathbb{R}^{n \times n}$ for the incomplete data $X_1^o \in \mathbb{R}^{d \times n}$ through similarity search applications, specifically maximum inner product search (MIPS) and nearest neighbor search (NNS). In these tasks, each incomplete sample in $X_1^o$ is treated as a query, and we perform one-vs-all retrieval, aiming to find the top-$N$ candidates with the highest inner products or smallest Euclidean distances. The search accuracy is measured using Recall, with Recall@N representing the average proportion of true top-$N$ results found within the top-$N$ retrieved candidates across all queries. For our experiments, we set $N = 10$, and refer to Recall@10 as Recall. A higher Recall indicates better preservation of local relationships (i.e., pairwise similarity or distance) across all samples.

**Table 2: Comparison of retrieval recall of maximum inner product search (MIPS) and nearest neighbor search (NNS) with $n = 1,000$ samples and $80\%$ random missing. Bold shows the best result, and underline marks the second-best.**

| Metric: Recall@10 ↑ | | Recall for MIPS | | | | Recall for NNS | | | |
|---|---|---|---|---|---|---|---|---|---|
| Baseline Type | Method | CIFAR10 | LFW | COIL100 | ISOLET | CIFAR10 | LFW | COIL100 | ISOLET |
| *Statistical* | Mean (2005) | $0.565_{\pm 0.010}$ | $0.541_{\pm 0.006}$ | $0.350_{\pm 0.005}$ | $0.234_{\pm 0.007}$ | $0.184_{\pm 0.011}$ | $0.187_{\pm 0.013}$ | $0.094_{\pm 0.008}$ | $0.077_{\pm 0.006}$ |
| *Imputation* | $k$NN (2016) | $0.604_{\pm 0.012}$ | $0.567_{\pm 0.008}$ | $0.371_{\pm 0.004}$ | $0.250_{\pm 0.007}$ | $0.186_{\pm 0.011}$ | $0.191_{\pm 0.015}$ | $0.114_{\pm 0.010}$ | $0.086_{\pm 0.009}$ |
| *Matrix* | SVT (2010) | $0.484_{\pm 0.009}$ | $0.438_{\pm 0.011}$ | $0.351_{\pm 0.005}$ | $0.215_{\pm 0.007}$ | $0.303_{\pm 0.012}$ | $0.285_{\pm 0.012}$ | $0.199_{\pm 0.003}$ | $0.138_{\pm 0.004}$ |
| *Completion* | KFMC (2019) | $0.602_{\pm 0.032}$ | $0.552_{\pm 0.006}$ | $0.261_{\pm 0.022}$ | $0.281_{\pm 0.007}$ | $0.205_{\pm 0.027}$ | $0.187_{\pm 0.013}$ | $0.157_{\pm 0.010}$ | $0.121_{\pm 0.008}$ |
| | PMC (2020) | $0.418_{\pm 0.035}$ | $0.573_{\pm 0.020}$ | $0.414_{\pm 0.024}$ | $0.227_{\pm 0.007}$ | $0.293_{\pm 0.014}$ | $0.248_{\pm 0.013}$ | $0.396_{\pm 0.014}$ | $0.251_{\pm 0.010}$ |
| *OT Imputation* | TDM (2023) | $0.275_{\pm 0.022}$ | $0.242_{\pm 0.031}$ | $0.248_{\pm 0.009}$ | $0.169_{\pm 0.012}$ | $0.152_{\pm 0.011}$ | $0.155_{\pm 0.011}$ | $0.088_{\pm 0.003}$ | $0.070_{\pm 0.007}$ |
| *Deep* | GAIN (2018) | $0.221_{\pm 0.088}$ | $0.286_{\pm 0.086}$ | $0.176_{\pm 0.077}$ | $0.241_{\pm 0.069}$ | $0.242_{\pm 0.026}$ | $0.275_{\pm 0.035}$ | $0.059_{\pm 0.015}$ | $0.086_{\pm 0.031}$ |
| *Imputation* | MIWAE (2019) | $0.258_{\pm 0.005}$ | $0.068_{\pm 0.021}$ | $0.226_{\pm 0.013}$ | $0.054_{\pm 0.011}$ | $0.086_{\pm 0.003}$ | $0.036_{\pm 0.002}$ | $0.077_{\pm 0.006}$ | $0.042_{\pm 0.006}$ |
| *Similarity* | DMC (2015) | $0.678_{\pm 0.004}$ | $0.656_{\pm 0.005}$ | $0.427_{\pm 0.005}$ | $0.281_{\pm 0.005}$ | $0.444_{\pm 0.009}$ | $0.515_{\pm 0.010}$ | $0.258_{\pm 0.008}$ | $0.192_{\pm 0.004}$ |
| *Calibration* | SMC (2023) | $\underline{0.719}_{\pm 0.004}$ | $\underline{0.692}_{\pm 0.006}$ | $\underline{0.455}_{\pm 0.006}$ | $\underline{0.310}_{\pm 0.006}$ | $0.472_{\pm 0.027}$ | $0.459_{\pm 0.024}$ | $0.355_{\pm 0.005}$ | $\underline{0.258}_{\pm 0.006}$ |
| | SVC (2024) | $0.630_{\pm 0.040}$ | $0.637_{\pm 0.021}$ | $0.423_{\pm 0.008}$ | $0.288_{\pm 0.008}$ | $0.505_{\pm 0.011}$ | $\underline{0.554}_{\pm 0.010}$ | $0.318_{\pm 0.012}$ | $0.221_{\pm 0.005}$ |
| *Initial Estimate* | $S^o$ & $D^o$ | $0.605_{\pm 0.005}$ | $0.582_{\pm 0.006}$ | $0.369_{\pm 0.004}$ | $0.236_{\pm 0.005}$ | $\underline{0.543}_{\pm 0.004}$ | $0.542_{\pm 0.006}$ | $\underline{0.380}_{\pm 0.006}$ | $0.224_{\pm 0.006}$ |
| *Our Method* | EC | $\mathbf{0.758}_{\pm 0.005}$ | $\mathbf{0.735}_{\pm 0.005}$ | $\mathbf{0.503}_{\pm 0.012}$ | $\mathbf{0.382}_{\pm 0.007}$ | $\mathbf{0.734}_{\pm 0.004}$ | $\mathbf{0.720}_{\pm 0.006}$ | $\mathbf{0.525}_{\pm 0.012}$ | $\mathbf{0.385}_{\pm 0.010}$ |
| | Improvement from $S^o$, $D^o$ to Ours | $25\%_{\pm 1\%}$ | $26\%_{\pm 1\%}$ | $36\%_{\pm 3\%}$ | $62\%_{\pm 3\%}$ | $35\%_{\pm 1\%}$ | $33\%_{\pm 1\%}$ | $38\%_{\pm 4\%}$ | $72\%_{\pm 5\%}$ |

As shown in Table 2, our method achieves the highest search accuracy for both MIPS and NNS, even with 80% missing data, maintaining recall scores above 0.7 for CIFAR10 and LFW datasets, effectively preserving pairwise relationships. In contrast, imputation methods perform poorly, particularly in NNS, as they fail to accurately recover pairwise distances and neighbor relationships. Similarly, calibration methods like DMC and SMC also show weak performance in NNS, often worse than the initial estimate $D^o$, consistent with the larger errors observed in Table 1.

## 6.4 Robustness Analysis

We assess the robustness by varying the missing rate $r$ from 20% to 80%. As shown in Fig. 9, our EC method consistently delivers accurate estimations ($\mathrm{RE}(S) < 0.16$, $\mathrm{RE}(D) < 0.05$) across all missing rates. Unlike imputation methods, which experience a sharp increase in RE and significant drops in recall, our method demonstrates minimal decline in performance for both MIPS and NNS. This highlights the robustness of our approach even under large missingness, with detailed numerical results provided in **Appendix D.1**.

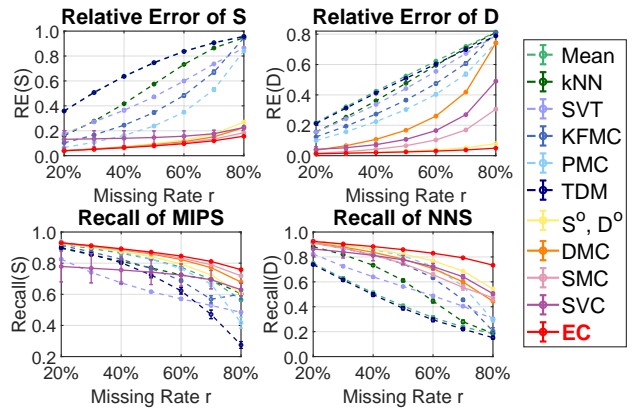

**Figure 9: Robustness analysis on the CIFAR10 with $n = 1,000$.**

## 6.5 Extension on Missing Mechanism

Our method described in Algorithm 2, which do not rely on the missing mechanism, adapts to various scenarios, including Missing at Random (**MAR**) [21] and Missing Not at Random (**MNAR**) [28]. Additionally, it accommodates more realistic missing patterns, such as **Segmental-Missing** (**SM**: missing in random length segments) and **Block-Missing** (**BM**: missing in random size blocks). Table 3 showcases the effectiveness across different mechanisms, where top-two lines denote the best performance achieved by imputation and calibration methods. Detailed results are in **Appendix D.5**.

**Table 3: Recall@10 of NNS task under various missing mechanisms on the CIFAR10 dataset with $n = 1,000$ and $r = 80\%$.**

| Mechanism | MAR | MNAR | SM | BM |
|---|---|---|---|---|
| Imputation | $0.607_{\pm 0.010}$ | $0.330_{\pm 0.013}$ | $0.310_{\pm 0.016}$ | $0.378_{\pm 0.014}$ |
| Calibration | $0.743_{\pm 0.012}$ | $0.523_{\pm 0.013}$ | $0.477_{\pm 0.013}$ | $0.359_{\pm 0.020}$ |
| EC (Ours) | $\mathbf{0.763}_{\pm 0.014}$ | $\mathbf{0.734}_{\pm 0.007}$ | $\mathbf{0.689}_{\pm 0.006}$ | $\mathbf{0.529}_{\pm 0.011}$ |

***Note.*** We provide comprehensive results of **ablation study** (Appendix D.1), **hyperparameter analysis** (Appendix D.2), **efficiency analysis** (Appendix D.3), **scalability analysis** (Appendix D.4), and the **extension on missing mechanism** (Appendix D.5).

## 7 Conclusion

Addressing the critical challenge of data incompleteness in inner product matrix estimation, we introduce a novel eigenvalue correction method. This method excels at reconstructing accurate inner product matrices from incomplete data by leveraging the Marchenko-Pastur Law. Unlike traditional imputation and calibration approaches, our method focuses on refining eigenvalue distributions to enhance accuracy in inner product and Euclidean distance estimations, thus improving similarity search tasks. Extensive experiments demonstrate our method's effectiveness and robustness in both maximum inner product search and nearest neighbor search tasks. Its adaptability to various missing mechanisms confirms its practical utility in real-world applications.

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

## A Proof

### A.1 Proof of Theorem 2

**Theorem 2 (Eigenvalue Distribution for Incomplete I.I.D. Data).** *Consider* $X^o = [x_1^o, \ldots, x_n^o] \in \mathbb{R}^{d \times n}$, *where the true values of* $\{x_{ij}^o\}$ *are i.i.d. random variables with mean 0 and variance* $\sigma^2 < \infty$, *missing completely at random (MCAR) with a missing rate of* $r \in (0, 1)$. *As* $d, n \to \infty$ *with* $d/n \to c \in (0, +\infty)$, *the limiting spectral distribution* $\mu^o$ *of the initial estimate* $S^o$ *is supported on*

$$[\lambda_-^o, \lambda_+^o] = \left[\frac{\sigma^2(1 - c^{-1/2})^2 - r}{1 - r}, \frac{\sigma^2(1 + c^{-1/2})^2 - r}{1 - r}\right]$$

*with the density function*

$$f^o(x) = \frac{c(1-r)^2}{2\pi\sigma^2} \frac{\sqrt{(\lambda_+^o - x)(x - \lambda_-^o)}}{(1-r)x + r} \mathbf{1}_{x \in [\lambda_-^o, \lambda_+^o]}.$$

PROOF. Given a complete version of the data matrix $X \in \mathbb{R}^{d \times n}$. We assume that each entry $x_{ij}$, for $1 \le i \le d$ and $1 \le j \le n$, can not be observed with probability $r_{ij} \in (0, 1)$, which is referred to as the *missing rate*. We consider *missing completely at random (MCAR)* with homogeneous missing, namely $r_{ij} = r$ for all $i \in [d]$ and $j \in [n]$. We introduce a matrix $B = [b_{ij}] \in \mathbb{R}^{d \times n}$ of *Bernoulli* variables to make this clear. Sample $b_{ij} \overset{i.i.d}{\sim}$ Bernoulli$(1 - r)$ independent of $X$, and we write

$$x_{ij}^o = b_{ij} x_{ij}, \ \forall i \in [d], j \in [n],$$

which is equivalent to the matrix form

$$X^o = B \circ X \in \mathbb{R}^{d \times n},$$

where $\circ$ denotes the *Hadamard product*. We should notice that

- $\mathbb{E}(x_{ij}) = 0$ and var$(x_{ij}) = \sigma^2$ for all $i \in [d]$ and $j \in [n]$;
- $\mathbb{E}(x_{ij}^o) = 0$ and var$(x_{ij}^o) = $ var$(b_{ij}) \cdot$ var$(x_{ij}) = (1-r)\sigma^2$ for all $i \in [d]$ and $j \in [n]$;
- $x_{ij}^o$ are independent of each other.

Then the LSD of $X^{o\top}X^o/d$ is MP$((1-r)\sigma^2, c^{-1})$. Here MP$(\sigma^2, c)$ corresponds to the LSD of sample covariance matrix with population covariance matrix $\Sigma = \sigma^2 I_d$ and aspect ratio $d/n \to c$.

The pairwise inner product matrix $S^o = [s_{ij}^o] \in \mathbb{R}^{n \times n}$ is calculated (here we assume the number of observed coordinates is approximately $(1-r)d$) as

$$s_{ij}^o = \begin{cases} \frac{1}{(1-r)^2 d} x_i^{o\top} x_j^o, & i \ne j, \\ \frac{1}{(1-r)d} x_i^{o\top} x_j^o, & i = j. \end{cases}$$

Writing this in a matrix form, we have

$$S^o = \frac{1}{(1-r)^2} X^{o\top}X^o/d - \frac{r}{(1-r)^2} \text{Diag}(X^{o\top}X^o/d),$$

where Diag$(X^{o\top}X^o)$ denotes the diagonal matrix with diagonal entries being those of $X^{o\top}X^o$. From the viewpoint of the law of large number, we have

$$\frac{1}{1-r} \text{Diag}(X^{o\top}X^o/d) \approx I_n, \quad S^o \approx \frac{1}{(1-r)^2} X^{o\top}X^o/d - \frac{r}{1-r} I_n,$$

where the approximation error caused by "≈" is $O_p(n^{-1/2} \log n)$ in the sense that $\|\text{Diag}(X^{o\top}X^o/d) - I_n\|_2 = O_p(n^{-1/2} \log n)$.

For the first matrix $\frac{1}{(1-r)^2} X^{o\top}X^o/d \in \mathbb{R}^{n \times n}$, we have

$$\begin{aligned} \text{Spec}\left\{\frac{1}{(1-r)^2} X^{o\top}X^o/d\right\} &= \frac{1}{(1-r)^2} \text{Spec}\{X^{o\top}X^o/d\} \\ &\to \frac{1}{(1-r)^2} \text{MP}((1-r)\sigma^2, c^{-1}), \end{aligned} \tag{A.1}$$

which is supported on $(1-r)^{-1}[\sigma^2(1 - c^{-1/2})^2, \sigma^2(1 + c^{-1/2})^2]$.

For the second matrix $\frac{r}{(1-r)^2}$Diag$(X^{o\top}X^o/d)$,

$$\frac{r}{(1-r)^2}\text{Diag}(X^{o\top}X^o/d) \approx \frac{r}{1-r} I_n.$$

Then the LSD of $S^o$ is supported on

$$\begin{aligned} &(1-r)^{-1}[\sigma^2(1 - c^{-1/2})^2, \sigma^2(1 + c^{-1/2})^2] - r(1-r)^{-1} \\ =& \left[\frac{\sigma^2(1 - c^{-1/2})^2 - r}{1 - r}, \frac{\sigma^2(1 + c^{-1/2})^2 - r}{1 - r}\right] \\ =&: [\lambda_-^o, \lambda_+^o]. \end{aligned} \tag{A.2}$$

From the Eq. (A.1), we can easily obtain the density function $f^o(x)$ of the LSD of $S^o$:

$$f^o(x) = \frac{c(1-r)^2}{2\pi\sigma^2 s} \frac{\sqrt{(\lambda_+^o - x)(x - \lambda_-^o)}}{(1-r)x + r} \mathbf{1}_{x \in [\lambda_-^o, \lambda_+^o]}. \tag{A.3}$$

□

### A.2 Proof of Theorem 3

**Theorem 3 (Optimality of Eigenvalue Correction Strategy).** *Given incomplete i.i.d. data* $X^o$ *with MCAR, the linear transformation* $\lambda_i^o \mapsto \hat{\lambda}_i := (1-r)\lambda_i^o + r$ *is the optimal transformation to reconstruct the spectral distribution of* $S^*$, *in the sense that almost surely* $|\hat{F}(x) - F^*(x)| \to 0$ *for any* $x \in \mathbb{R}$, *where* $\hat{F}(x)$ *and* $F^*(x)$ *are distribution functions corresponding to* $\{\hat{\lambda}_i\}$ *and* $\{\lambda_i^*\}$, *respectively.*

PROOF. On one hand, Lemma 1 implies that almost surely

$$|F^*(x) - \mu^*(x)| \to 0, \ \forall x \in \mathbb{R}, \tag{A.4}$$

where $\mu^*$ equals to the MP law $\mu_c$ with aspect ratio $c$. On the other hand, Theorem 2 implies that almost surely

$$|F^o(x) - \mu^o(x)| \to 0 \tag{A.5}$$

and

$$\mu^o(x) = \mu_c((1-r)x + r) \tag{A.6}$$

for all $x \in \mathbb{R}$ under MCAR, where $F^o$ and $\mu^o$ denote the ESD and LSD of $S^o$, respectively. Thus, from Eqs. (A.4), (A.5) and (A.6), we conclude that almost surely

$$|\hat{F}(x) - F^*(x)| \to 0, \ \forall x \in \mathbb{R}, \tag{A.7}$$

since the linear transformation $\lambda_i^o \mapsto \hat{\lambda}_i := (1-r)\lambda_i^o + r$ leads to $\hat{F}(x) = F^o((1-r)^{-1}(x - r))$.

□

## A.3 Proof of Theorem 4

**Theorem 4 (Error Bound of Inner Product Estimation).** *Given incomplete i.i.d. data $X^o$ with MCAR, for any small constant $\varepsilon$, it holds with probability $(1 - o(1))$ that $\|\hat{S} - S^*\|_F \leq (\eta_S + \varepsilon)\|S^o - S^*\|_F$, where $\eta_S = \sqrt{1 - \frac{r^2 c^{-1}}{(2 + c^{-1})(1 - r)^2 + 2r(1 - r) + c^{-1}}} \in (0, 1)$.*

PROOF. First, we have

$$\|\hat{S} - S^*\|_F^2 = \mathrm{tr}\left[(\hat{S} - S^*)(\hat{S} - S^*)^\top\right] = \mathrm{tr}(\hat{S}^2) - 2\mathrm{tr}(S^*\hat{S}) + \mathrm{tr}(S^{*2})$$

and

$$\|S^o - S^*\|_F^2 = \mathrm{tr}\left[(S^o - S^*)(S^o - S^*)^\top\right] = \mathrm{tr}(S^{o2}) - 2\mathrm{tr}(S^*S^o) + \mathrm{tr}(S^{*2})$$

It follows that

$$\|S^o - S^*\|_F^2 - \|\hat{S} - S^*\|_F^2 = \mathrm{tr}(S^{o2}) - \mathrm{tr}(\hat{S}^2) + 2\mathrm{tr}\left[S^*(\hat{S} - S^o)\right]$$

Note that $\hat{S} = (1 - r)S^o + rI_n$ due to the linear transformation $\hat{\lambda}_i = (1 - r)\lambda_i^o + r$ in Theorem 3 for $1 \leq i \leq n$. It leads to

$$\hat{S} - S^o = -rS^o + rI_n$$
$$\hat{S}^2 = (1 - r)^2 S^{o2} + 2r(1 - r)S^o + r^2 I_n$$

and then

$$\mathrm{tr}\left[S^*(\hat{S} - S^o)\right] = -r\mathrm{tr}\left(S^*S^o\right) + r\mathrm{tr}(S^*),$$
$$\mathrm{tr}(\hat{S}^2) = (1 - r)^2\mathrm{tr}(S^{o2}) + 2r(1 - r)\mathrm{tr}(S^o) + r^2 n.$$

On one hand, it holds with high probability, i.e., probability $(1 - o(1))$, that

$$\frac{1}{n}\left(\|S^o - S^*\|_F^2 - \|\hat{S} - S^*\|_F^2\right)$$
$$= \frac{1}{n}\left\{\mathrm{tr}(S^{o2}) - \mathrm{tr}(\hat{S}^2) + 2\mathrm{tr}\left[S^*(\hat{S} - S^o)\right]\right\}$$
$$= \frac{1}{n}\left\{(2r - r^2)\mathrm{tr}(S^{o2}) - 2r(1 - r)\mathrm{tr}(S^o) - r^2 - 2r\mathrm{tr}(S^*S^o) + 2r\mathrm{tr}(S^*)\right\}$$
$$\geq (2r - r^2)\left(1 + \frac{c^{-1}}{(1 - r)^2}\right) - 2r(1 - r) - r^2 - 2r\left(1 + \frac{c^{-1}}{1 - r}\right) + 2r - \varepsilon$$
$$= \frac{r^2 c^{-1}}{(1 - r)^2} - \varepsilon \tag{A.7}$$

for any small constant $\varepsilon > 0$, since

$$\frac{1}{n}\mathrm{tr}(S^o) = \frac{1}{n}\mathrm{tr}\left(\frac{1}{(1 - r)^2}X^{o\top}X^o/d - \frac{r}{(1 - r)^2}\mathrm{Diag}(X^{o\top}X^o/d)\right)$$
$$\xrightarrow{p} (1 - r)^{-1} - r(1 - r)^{-1} = 1,$$

$$\frac{1}{n}\mathrm{tr}(S^{o2}) = \frac{1}{n}\mathrm{tr}\left\{\left(\frac{1}{(1 - r)^2}X^{o\top}X^o/d - \frac{r}{(1 - r)^2}\mathrm{Diag}(X^{o\top}X^o/d)\right)^2\right\}$$
$$= \frac{1}{n}\mathrm{tr}\left\{\left(\frac{1}{(1 - r)^2}X^{o\top}X^o/d\right)^2\right\}$$
$$\quad - \frac{2}{n}\mathrm{tr}\left\{\left(\frac{1}{(1 - r)^2}X^{o\top}X^o/d\right)\mathrm{Diag}\left(\frac{r}{(1 - r)^2}X^{o\top}X^o/d\right)\right\}$$
$$\quad + \frac{1}{n}\mathrm{tr}\left\{\mathrm{Diag}\left(\frac{1}{(1 - r)^2}X^{o\top}X^o/d\right)^2\right\}$$
$$\xrightarrow{p} \frac{1 + c^{-1}}{(1 - r)^2} - \frac{2r}{(1 - r)^2} + \frac{r^2}{(1 - r)^2}$$
$$= 1 + \frac{c^{-1}}{(1 - r)^2},$$

$$\frac{1}{n}\mathrm{tr}(S^*S^o) = \frac{1}{1 - r}\frac{1}{n}\mathrm{tr}(S^*\hat{S}) - \frac{r}{1 - r}$$
$$\leq \frac{1}{1 - r}\sqrt{\frac{1}{n}\mathrm{tr}(S^{*2}) \cdot \frac{1}{n}\mathrm{tr}(\hat{S}^2)} - \frac{r}{1 - r}$$
$$\xrightarrow{p} \frac{1}{1 - r}\sqrt{(1 + c^{-1})(1 + c^{-1})} - \frac{r}{1 - r}$$
$$= 1 + \frac{c^{-1}}{1 - r}.$$

On the other hand, it holds that

$$\frac{1}{n}\|S^o - S^*\|_F^2 = \frac{1}{n}\mathrm{tr}(S^{o2}) - \frac{2}{n}\mathrm{tr}(S^*S^o) + \frac{1}{n}\mathrm{tr}(S^{*2})$$
$$= \frac{1}{n}\mathrm{tr}(S^{o2}) - 2\left(\frac{1}{1 - r}\frac{1}{n}\mathrm{tr}(S^*\hat{S}) - \frac{r}{1 - r}\right) + \frac{1}{n}\mathrm{tr}(S^{*2})$$
$$\leq \frac{1}{n}\mathrm{tr}(S^{o2}) + \frac{2r}{1 - r} + \frac{1}{n}\mathrm{tr}(S^{*2})$$
$$\xrightarrow{p} 1 + \frac{c^{-1}}{(1 - r)^2} + \frac{2r}{1 - r} + (1 + c^{-1})$$
$$= 2 + c^{-1} + \frac{c^{-1}}{(1 - r)^2} + \frac{2r}{1 - r}, \tag{A.8}$$

where in the third step we used the fact that $\mathrm{tr}(S^*\hat{S}) \geq 0$ since both $S^*$ and $\hat{S}$ are non-negative definite.

Thus, we can conclude from Eq. (A.7) and Eq. (A.8) that

$$\frac{\|S^o - S^*\|_F^2 - \|\hat{S} - S^*\|_F^2}{\|S^o - S^*\|_F^2} \geq \frac{\frac{r^2 c^{-1}}{(1 - r)^2}}{2 + c^{-1} + \frac{c^{-1}}{(1 - r)^2} + \frac{2r}{1 - r}} - \varepsilon$$
$$= \frac{r^2 c^{-1}}{(2 + c^{-1})(1 - r)^2 + 2r(1 - r) + c^{-1}} - \varepsilon$$

holds with high probability for any small constant $\varepsilon > 0$. Equivalently, we take

$$\eta_S = \sqrt{1 - \frac{r^2 c^{-1}}{(2 + c^{-1})(1 - r)^2 + 2r(1 - r) + c^{-1}}},$$

then for any small constant $\varepsilon > 0$, it holds with high probability:

$$\frac{\|\hat{S} - S^*\|_F}{\|S^o - S^*\|_F} \leq \eta_S + \varepsilon.$$

Finally, it is not hard to verify that $\eta_S \in (0, 1)$ when $r \in (0, 1)$ and $c \in (0, \infty)$.

$\square$

## A.4 Proof of Theorem 5

**Theorem 5 (Eigenvalue Distribution for Incomplete Separable Data).** *Consider non-i.i.d. separable data* $X = [x_1, \ldots, x_n] \in \mathbb{R}^{d \times n}$, *where* $x_i = \Sigma^{1/2} z_i \in \mathbb{R}^d$, *with* $z_i$ *having independent coordinates,* $\mathbb{E}[z_i] = 0$, *and* $\mathrm{Cov}(z_i) = I_d$. *Define* $X^o$ *as the incomplete version of* $X$ *with MCAR in a missing rate* $r$, *and* $S^o$ *as the initial inner product matrix of* $X^o$. *For the eigenvalues* $\{\lambda_i^o\}$ *of* $S^o$, *it holds that, for* $1 \le i \le n$,

$$\lambda_i^o - (1-r)^{-1}\lambda_i^* \xrightarrow{p} r(1-r)^{-1}\mathrm{tr}(\Sigma)/d,$$

*where* $\xrightarrow{p}$ *indicates convergence in probability and* $\lambda_i^*$ *is the i-th eigenvalue of ground-truth* $S^*$.

PROOF. Consider the observed data matrix $X_n^o = (x_1^o, \cdots, x_n^o) \in \mathbb{R}^{d \times n}$ with ground truth $X_n = (x_1, \cdots, x_n) \in \mathbb{R}^{d \times n}$. Suppose $x_i = \Sigma^{1/2} z_i$. Then it holds that

$$x_{ij}^o = b_{ij} \cdot x_{ij} = b_{ij} z_i^\top \Sigma_j^{1/2}, \ i \in [d], \ j \in [n],$$

where $x_i = \Sigma^{1/2} z_i$ and $b_{ij} \overset{i.i.d}{\sim} \mathrm{Bernoulli}(1-r)$ and $\Sigma_j^{1/2}$ denotes the $j$-th column of $\Sigma^{1/2}$. Since $x_i$ follows Gaussian distribution, then $x_i \overset{d}{=} U x_i$ for any $d \times d$ orthogonal matrix $U$. So it suffices to deal with the simple case of diagonal $\Sigma$, that is, $\Sigma_{ij} = 0$ for any $i \ne j$. We denote $\Sigma = \mathrm{Diag}(\sigma_1^2, \cdots, \sigma_d^2) \in \mathbb{R}^{d \times d}$. It follows that

$$x_{ij}^o = b_{ij}\sigma_j z_{ij},$$

or equivalently,

$$x_i^o = \Sigma^{1/2}(B_i \circ z_i) =: \Sigma^{1/2} z_i^o, \tag{A.9}$$

where we use $z_i^o := B_i \circ z_i$ to denote the counterpart of $x_i^o$ for $i \in [n]$. It is not hard to verify that

$$\mathbb{E}(z_{ij}^o) = 0, \ \mathrm{var}(z_{ij}^o) = 1 - r, \ z_{i_1 j_1}^o \perp z_{i_2 j_2}^o,$$

$\forall (i, j) \in [n] \times [d]$ and $(i_1, j_1) \ne (i_2, j_2) \in [n] \times [d]$.

This leads to that

$$\mathrm{Spec}(X_n^{o\top} X_n^o / d) = (1-r) \cdot \mathrm{Spec}(X_n^\top X_n / d). \tag{A.10}$$

Recalling the definition of inner product matrix $S_n^o$, we have

$$S_n^o = \frac{1}{(1-r)^2} X_n^{o\top} X_n^o / d - \frac{r}{(1-r)^2} \mathrm{Diag}(X_n^{o\top} X_n^o / d) \tag{A.11}$$

For the second term on the right hand side, we have

$$\mathrm{Diag}(X_n^{o\top} X_n^o / d) \approx (1-r)\mathrm{tr}(\Sigma)/d \cdot I_n \tag{A.12}$$

since $x_i^{o\top} x_i^o / d = z_i^{o\top} \Sigma z_i^o / d \to (1-r)\mathrm{tr}(\Sigma)/d$ almost surely for all $1 \le i \le n$. More specifically, we have

$$\left\| \mathrm{Diag}(X_n^{o\top} X_n^o / d) - (1-r)\mathrm{tr}(\Sigma)/d \cdot I_n \right\|_2 = O_p(n^{-1/2}\log n).$$

Thus, we can conclude that

$$\lambda_i(S_n^o) - (1-r)^{-1}\mathrm{supp}(X_n^\top X_n / d) \xrightarrow{p} r(1-r)^{-1}\mathrm{tr}(\Sigma)/d$$

for $1 \le i \le n$. Also, (A.10), (A.11) and (A.12) together imply that the LSD of $S^o$ and $X_n^\top X_n / d$ share the same "shape". □

## A.5 Proof of Theorem 6

PROOF. Theorem 6 can be directly implied from the MP Law [19]. □

## A.6 Proof of Theorem 7

**Theorem 7 (Error Bound of Euclidean Distance Estimation).** *Given incomplete i.i.d data* $X^o$ *with MCAR, there exists* $\eta_D \in (0, 1)$ *such that* $\|\hat{D} - D^*\|_F \le (\eta_D + \varepsilon)\|D^o - D^*\|_F$ *holds with probability* $(1 - o(1))$ *for any small* $\varepsilon > 0$, *with* $\eta_D$ *specified in Eq. (A.13).*

PROOF. First, by the definition of Frobenius norm, we have

$$\|\hat{D} - D^*\|_F^2 = \mathrm{tr}\left[(\hat{D} - D^*)(\hat{D} - D^*)^\top\right] = \mathrm{tr}(\hat{D}^2) - 2\mathrm{tr}(\hat{D}D^*) + \mathrm{tr}(D^{*2})$$

and

$$\|D^o - D^*\|_F^2 = \mathrm{tr}\left[(D^o - D^*)(D^o - D^*)^\top\right]$$
$$= \mathrm{tr}(D^{o2}) - 2\mathrm{tr}(D^o D^*) + \mathrm{tr}(D^{*2}).$$

This leads to

$$\|D^o - D^*\|_F^2 - \|\hat{D} - D^*\|_F^2 = \mathrm{tr}(D^{o2}) - 2\mathrm{tr}(D^o D^*) - \mathrm{tr}(\hat{D}^2) + 2\mathrm{tr}(\hat{D}D^*)$$

Recall Eq. (3) that

$$\hat{D} = \mathrm{Diag}(d\hat{S}) \cdot J + J \cdot \mathrm{Diag}(d\hat{S}) - 2d\hat{S}.$$

Also, we have

$$D^o = \mathrm{Diag}(dS^o) \cdot J + J \cdot \mathrm{Diag}(dS^o) - 2dS^o,$$
$$D^* = \mathrm{Diag}(dS^*) \cdot J + J \cdot \mathrm{Diag}(dS^*) - 2dS^*.$$

Using the approximations $\| \mathrm{Diag}(\hat{S}) - I_n \|_2 = O_p(n^{-1/2}\log n)$, $\| \mathrm{Diag}(S^o) - I_n \|_2 = O_p(n^{-1/2}\log n)$, $\| \mathrm{Diag}(S^*) - I_n \|_2 = O_p(n^{-1/2}\log n)$, we have

$$\hat{D}/d \approx 2J - 2\hat{S}, \ D^o/d \approx 2J - 2S^o, \ D^*/d \approx 2J - 2S^*,$$

where the approximation error (in $\ell_2$ norm) is $o_p(1)$. It follows that

$$\hat{D}^2/d^2 \approx 4(J^2 - J\hat{S} - \hat{S}J + \hat{S}^2),$$
$$D^{o2}/d^2 \approx 4(J^2 - JS^o - S^o J + S^{o2}),$$
$$D^{*2}/d^2 \approx 4(J^2 - JS^* - S^* J + S^{*2})$$

and

$$D^o D^*/d^2 \approx 4(J^2 - JS^* - S^o J + S^o S^*),$$
$$\hat{D}D^*/d^2 \approx 4(J^2 - JS^* - \hat{S}J + \hat{S}S^*).$$

Then we have

$$\mathrm{tr}(D^{o2})/d^2 \approx 4(\mathrm{tr}(J^2) - 2\mathrm{tr}(JS^o) + \mathrm{tr}(S^{o2}))$$
$$= 4n^2 - 8\mathrm{tr}\{J[(1-r)^{-1}(\hat{S} - rI_n)]\} + 4\mathrm{tr}(S^{o2}),$$
$$\approx 4n^2 + 8r(1-r)^{-1}n - 8(1-r)^{-1}\mathrm{tr}(J\hat{S}) + 4n\left[1 + c^{-1}(1-r)^{-2}\right]$$
$$\mathrm{tr}(D^o D^*)/d^2 \approx 4(\mathrm{tr}(J^2) - \mathrm{tr}(JS^*) - \mathrm{tr}(S^o J) + \mathrm{tr}(S^o S^*))$$
$$= 4n^2 - 4\mathrm{tr}(JS^*) - 4\mathrm{tr}(S^o J) + 4\mathrm{tr}(S^o S^*),$$
$$\approx 4n^2 - 4\mathrm{tr}(JS^*) - 4(1-r)^{-1}\mathrm{tr}(\hat{S}J) + 4r(1-r)^{-1}n$$
$$+ 4(1-r)^{-1}\mathrm{tr}(S^o S^*) - 4r(1-r)^{-1}n$$
$$\approx 4n^2 - 4\mathrm{tr}(JS^*) - 4(1-r)^{-1}\mathrm{tr}(\hat{S}J) + 4(1-r)^{-1}\mathrm{tr}(\hat{S}S^*)$$
$$\mathrm{tr}(\hat{D}^2)/d^2 \approx 4(\mathrm{tr}(J^2) - 2\mathrm{tr}(J\hat{S}) + \mathrm{tr}(\hat{S}^2))$$
$$\approx 4n^2 - 8\mathrm{tr}(J\hat{S}) + 4n\left(1 + c^{-1}\right),$$
$$\mathrm{tr}(\hat{D}D^*)/d^2 \approx 4(\mathrm{tr}(J^2) - \mathrm{tr}(JS^*) - \mathrm{tr}(\hat{S}J) + \mathrm{tr}(\hat{S}S^*))$$
$$= 4n^2 - 4\mathrm{tr}(JS^*) - 4\mathrm{tr}(\hat{S}J) + 4\mathrm{tr}(\hat{S}S^*),$$

where all approximations "$\approx$" hold in the sense of convergence in probability (after appropriate scaling). Thus, it holds that

$$(\|D^o - D^*\|_F^2 - \|\hat{D} - D^*\|_F^2)/d^2$$

$$= (\text{tr}(D^{o2}) - 2\text{tr}(D^o D^*) - \text{tr}(\hat{D}^2) + 2\text{tr}(\hat{D}D^*))/d^2$$

$$\approx 8r(1-r)^{-1}n + 4c^{-1}r(2-r)(1-r)^{-2}n - 8r(1-r)^{-1}\text{tr}(\hat{S}S^*)$$

$$\geq 8r(1-r)^{-1}n + 4c^{-1}r(2-r)(1-r)^{-2}n - 8r(1-r)^{-1}(1+c^{-1})n - \varepsilon n$$

with probability $(1 - o(1))$ for any constant $\varepsilon > 0$, since $\text{tr}(\hat{S}S^*) \leq \sqrt{\text{tr}(\hat{S}^2)\text{tr}(S^{*2})} \approx (1 + c^{-1})n$. Also, it holds with probability $(1 - o(1))$ that

$$\|D^o - D^*\|_F^2/d^2$$

$$= \left(\text{tr}(D^{o2}) - 2\text{tr}(D^o D^*) + \text{tr}(D^{*2})\right)/d^2$$

$$\approx 8r(1-r)^{-1}n + 4(2 + c^{-1} + c^{-1}(1-r)^{-2})n - 8(1-r)^{-1}\text{tr}(\hat{S}S^*)$$

$$\leq 8r(1-r)^{-1}n + 4(2 + c^{-1} + c^{-1}(1-r)^{-2})n + \varepsilon n$$

for any constant $\varepsilon > 0$. Taking

$$\eta_D = \sqrt{1 - \frac{8r(1-r)^{-1} + 4c^{-1}r(2-r)(1-r)^{-2} - 8r(1-r)^{-1}(1+c^{-1})}{8r(1-r)^{-1} + 4(2 + c^{-1} + c^{-1}(1-r)^{-2})}},$$
(A.13)

then for any small constant $\varepsilon > 0$, it holds with probability $(1 - o(1))$ that

$$\frac{\|\hat{D} - D^*\|_F}{\|D^o - D^*\|_F} \leq \eta_D + \varepsilon.$$

It can be verified that $\eta_D \in (0, 1)$ when $r \in (0, 1)$ and $c \in (0, \infty)$.

$\square$

## B  Algorithm

### B.1  Scalable Algorithm for Non-I.I.D. Data

Consider a large dataset with unequal-sized subsets: an incomplete $X_1^o \in \mathbb{R}^{d \times 10,000}$ and a complete $X_2 \in \mathbb{R}^{d \times 5,000}$. We partition the similarity matrices $S^o$ and $S$ into smaller submatrices to apply a divide-and-conquer strategy. Let $m = 2,500$, so $S^o \in \mathbb{R}^{10,000 \times 10,000}$ is split into 16 submatrices and $S \in \mathbb{R}^{5,000 \times 5,000}$ into 4 submatrices, each of size $2,500 \times 2,500$, as shown below:

$$S^o = \begin{bmatrix} S_{11}^o & S_{12}^o & S_{13}^o & S_{14}^o \\ S_{21}^o & S_{22}^o & S_{23}^o & S_{24}^o \\ S_{31}^o & S_{32}^o & S_{33}^o & S_{34}^o \\ S_{41}^o & S_{42}^o & S_{43}^o & S_{44}^o \end{bmatrix} \quad S = \begin{bmatrix} S_{11} & S_{12} \\ S_{21} & S_{22} \end{bmatrix},$$

where **bold** represents the diagonal blocks. The procedure of scalable eigenvalue correction is as follows.

**1. Correcting Diagonal Submatrices:**

- Perform eigen-decomposition on $S_{11}$ and $S_{22}$ as $S_{ii} = U_{ii}^* \Lambda_{ii}^* U_{ii}^{*\top}$, and average the eigenvalues: $\Lambda^* = (\Lambda_{11}^* + \Lambda_{22}^*)/2$.
- For each $S_{ii}^o$, perform eigen-decomposition $S_{ii}^o = U_{ii}^o \Lambda_{ii}^o U_{ii}^{o\top}$, and replace the small eigenvalues of $\Lambda_{ii}^o$ with those of $\Lambda^*$ to obtain $\hat{\Lambda}_{ii}$.
- Reconstruct the corrected diagonal submatrices: $\hat{S}_{ii} = U_{ii}^o \hat{\Lambda}_{ii} U_{ii}^{o\top}$.

**2. Correcting Off-diagonal Submatrices:**

- Perform singular value decomposition (SVD) on $S_{12}^*$ and $S_{21}^*$, and average the singular values: $\Sigma^* = (\Sigma_{12}^* + \Sigma_{21}^*)/2$.
- For each off-diagonal submatrix $S_{ij}^o$, perform SVD: $S_{ij}^o = U_{ij}^o \Sigma_{ij}^o V_{ij}^{o\top}$, and update $\Sigma_{ij}^o$ by replacing small singular values with those from $\Sigma^*$.
- Reconstruct the corrected off-diagonal submatrices: $\hat{S}_{ij} = U_{ij}^o \hat{\Sigma}_{ij} V_{ij}^{o\top}$.

This approach yields an enhanced similarity matrix $\hat{S} = (\hat{S}_{ij})$ for the incomplete data $X_1^o$ via a divide-and-conquer strategy, where the complete steps are summarized in Algorithm 3.

---

**Algorithm 3 Scalable Eigenvalue Correction for Non-I.I.D. Data**

---

**Input:** $X_1^o \in \mathbb{R}^{d \times n_1}$: an incomplete subset; $X_2 \in \mathbb{R}^{d \times n_2}$: a complete subset; $k$: top-$k$ eigenvalues or singular values (hyperparameter); $m$: the partition size (hyperparameter).

**Output:** $\hat{S} \in \mathbb{R}^{n_1 \times n_1}$: the corrected inner product matrix for $X_1^o$.

1: Set $N_1 = n_1/m$ and $N_2 = n_2/m$.
2: Calculate $S^o \in \mathbb{R}^{n_1 \times n_1}, S \in \mathbb{R}^{n_2 \times n_2}$ from $X_1^o, X_2$ via Eq. (1).
3: Partition $S^o$ into submatrices $\{S_{ij}^o\}_{i,j=1}^{N_1}$ of size $m \times m$;
4: Partition $S$ into submatrices $\{S_{pq}\}_{p,q=1}^{N_2}$ of size $m \times m$.
5: ▷ *Stage-I. Correcting Diagonal Submatrices*
6: **parfor** $p = 1, 2, \ldots, N_2$ **do**
7:     Perform eigen-decomposition: $S_{pp} = U_{pp} \Lambda_{pp}^* U_{pp}^\top$;
8: **end**
9: Calculate average eigenvalues in $S$: $\Lambda^* = \frac{1}{N_2}\sum_{p=1}^{N_2} \Lambda_{pp}^* = \text{Diag}(\lambda_1^*, \lambda_2^*, \ldots, \lambda_m^*)$.
10: **parfor** $i = 1, 2, \ldots, N_1$ **do**
11:     Perform eigen-decomposition: $S_{ii}^o = U_{ii}^o \Lambda_{ii}^o U_{ii}^{o\top}$ with $\Lambda_{ii}^o = \text{Diag}(\lambda_1^o, \lambda_2^o, \ldots, \lambda_m^o)$;
12:     Correct the eigenvalues by $\hat{\Lambda}_{ii} = \text{Diag}(\underbrace{\lambda_1^o, \cdots, \lambda_k^o}_{\text{from } S^o}, \underbrace{\lambda_{k+1}^*, \cdots, \lambda_m^*}_{\text{from } S})$;
13:     Obtain corrected diagonal submatrix: $\hat{S}_{ii} = U_{ii}^o \hat{\Lambda}_{ii} U_{ii}^{o\top}$.
14: **end**
15: ▷ *Stage-II. Correcting Off-diagonal Submatrices*
16: **parfor** $p, q = 1, 2, \ldots, N_2$ ($p \neq q$) **do**
17:     Perform singular value decomposition: $S_{pq} = U_{pq} \Sigma_{pq}^* V_{pq}^\top$;
18: **end**
19: Calculate average singular values in $S$: $\Sigma^* = \frac{1}{N_2(N_2-1)}\sum_{p \neq q} \Sigma_{pq}^* = \text{Diag}(\sigma_1^*, \sigma_2^*, \ldots, \sigma_m^*)$.
20: **parfor** $i, j = 1, 2, \ldots, N_1$ ($i \neq j$) **do**
21:     Perform singular value decomposition: $S_{ij}^o = U_{ij}^o \Sigma_{ij}^o V_{ij}^{o\top}$ with $\Sigma_{ij}^o = \text{Diag}(\sigma_1^o, \sigma_2^o, \ldots, \sigma_m^o)$;
22:     Correct singular values by $\hat{\Sigma}_{ij} = \text{Diag}(\underbrace{\sigma_1^o, \cdots, \sigma_k^o}_{\text{from } S^o}, \underbrace{\sigma_{k+1}^*, \cdots, \sigma_m^*}_{\text{from } S})$;
23:     Obtain corrected off-diagonal submatrix: $\hat{S}_{ij} = U_{ij}^o \hat{\Sigma}_{ij} V_{ij}^{o\top}$.
24: **end**
25: **Return** $\hat{S} = (\hat{S}_{ij}) \in \mathbb{R}^{n_1 \times n_1}$.

---

**Quadratic Time Complexity.** Given a partition size of $m$, $S^o \in \mathbb{R}^{n_1 \times n_1}$ is divided into $\frac{n_1^2}{m^2}$ submatrices of size $m \times m$. Similarly,

$S \in \mathbb{R}^{n_2 \times n_2}$ is divided into $\frac{n_2^2}{m^2}$ submatrices. In total, there are $\frac{n_1^2 + n_2^2}{m^2}$ submatrices. The time complexity for the eigen-decomposition or singular value decomposition of each $m \times m$ submatrix is $O(m^3)$. Therefore, the overall time complexity of Algorithm 3 is $\frac{n_1^2 + n_2^2}{m^2} \cdot O(m^3) = O(mn_1^2 + mn_2^2)$, resulting in **quadratic complexity** and a significant reduction in running time. Additionally, all loops (Lines 6-8, 10-14, 16-18, and 20-24) are designed for parallel execution, further enhancing efficiency.

## C  Experimental Settings

### C.1  Datasets

We utilize four well-known benchmark datasets that cover a reasonable range of application domains, encompassing various types of images and speech data.

- **CIFAR10** [15] [1]: a color-image dataset consists of 60,000 color images of $32 \times 32$ pixels across 10 classes, each image reshaped into a 3,072-dimensional vector ($d = 3,072$).
- **LFW** [12] [2]: a face-image dataset features 13,233 images of faces, each resized to $64 \times 64$ pixels in grayscale and reshaped into a 4,096-dimensional vector ($d = 4,096$);
- **COIL100** [23] [3]: an object-image dataset comprises 7,200 object images in 100 classes, each resized to $32 \times 32$ pixels in grayscale and reshaped into a 1,024-dimensional vector ($d = 1,024$).
- **ISOLET** [5] [4]: a speech dataset contains 7,797 recordings of different speakers, each represented by a 617-dimensional vector ($d = 617$).

### C.2  Baseline Methods and Hyperparameters

Our approach is evaluated against a range of representative methods designed to incomplete data:

- **Mean** [11]: Replaces missing values with the mean of observed values in the corresponding feature.
- **$k$NN** [1]: Imputes missing values using average values of $k$-nearest neighbors (default: $k = 10$).
- **SVT** [3]: Employs singular value thresholding for low-rank matrix completion.
- **KFMC** [6]: Utilizes a kernelized factorization technique for high-rank matrix completion in the offline pattern (default: polynomial kernel).
- **PMC** [7]: Applies polynomial matrix completion for low-rank matrix completion (default: polynomial kernel).
- **TDM** [28]: Uses transformed distribution matching for optimal-transport-based imputation, requiring more than 6 hours for 1,000 iterations (default: $T = 3$ and $K = 2$ for 1,000 iterations).
- **GAIN** [25]: Uses generative adversarial nets (GAN) framework to impute the missing components conditioned on what is actually observed.
- **MIWAE** [20]: Uses the importance-weighted autoencoder and maximises a potentially tight lower bound of the log-likelihood of the observed data.

---

[1] https://www.cs.toronto.edu/~kriz/cifar.html
[2] https://vis-www.cs.umass.edu/lfw/
[3] https://www1.cs.columbia.edu/CAVE/software/softlib/coil-100.php
[4] http://archive.ics.uci.edu/ml/datasets/ISOLET

- **DMC** [16]: Adjusts an initial similarity matrix $S^o$ to its nearest positive semi-definite (PSD) matrix through convex optimization, specifically by solving $\min_{S \succeq 0} \|S - S^o\|_F^2$. This is achieved by setting all negative eigenvalues of the inner product matrix to zero.
- **SMC** [26]: Calibrates an initial similarity matrix $S^o$ towards PSD by sequentially updating the similarity vector $v$ to solve $\min_v \|S - S^o\|_F^2$ subject to $S \succeq 0$.
- **SVC** [18]: Adjusts an initial similarity matrix $S^o$ to the PSD matrix by batch calibrating similarity vectors. The optimization involves solving $\min_v \frac{1}{2}(v - v^o)^\top (v - v^o)$ under constraint $v^\top S^{-1} v \leq 1$, where $v$ is the similarity vector.

### C.3  Implementation Details

**Implementation.**  In all experiments, we randomly select $n$ samples to form the incomplete dataset $X_1^o \in \mathbb{R}^{d \times n}$, with entries missing according to different mechanisms, and another $n$ samples for the complete dataset $X_2 \in \mathbb{R}^{d \times n}$. These are combined into a single data matrix $X = [X_1^o, X_2]$, used by all imputation algorithms to generate an imputed matrix $\hat{X} = [\hat{X}_1, X_2]$. The inner product matrix $\hat{S}_1$ is then computed from $\hat{X}_1$ for imputation methods. For calibration methods, the process begins with the initial inner product matrix $S_1^o$ from $X_1^o$, as defined by Eq. (1), which is optimized to $\hat{S}_1$ using various optimization techniques. In short, the route of imputation is $X_1^o \rightarrow \hat{X}_1 \rightarrow \hat{S}_1$, while the route of calibration is $X_1^o \rightarrow S_1^o \rightarrow \hat{S}_1$. Finally, all experiments of estimation errors and similarity search performance are conducted on $\hat{S}_1$.

**Similarity Search Tasks.**  We perform one-versus-all similarity searches. In the maximum inner product search, each sample in the incomplete dataset $X_1^o$ serves as a query. Using the estimated inner product matrix $\hat{S}_1$, we identify the top-$N$ candidates with the highest inner product. The Recall@N for each sample is calculated as the proportion of true top-$N$ items among the top-$N$ candidates. The average Recall@N is then recorded across all samples. Similarly, the nearest neighbor search involves identifying the top-$N$ candidates with the smallest Euclidean distance for each query.

## D  Comprehensive Results and Analysis

To demonstrate the effectiveness of our method, we provide comprehensive results in this section, including ablation study (Section D.1), hyperparameter analysis (Section D.2), efficiency analysis (Section D.3), and scalability analysis (Section D.4), followed by extension on various missing mechanisms (Section D.5).

### D.1  Ablation Study

As shown in Table D.4, even with a small missing rate (e.g., 20%), while the initial estimate $S^o$ ($D^o$) is close to the ground-truth, our EC method enhances it by reducing relative errors and improving search accuracy. As missing rates increase, our EC method continues to demonstrate stable search accuracy, confirming its consistent improvements from the initial estimate across various levels of missing data.

**Table D.4: Ablation study under various missing rates on the incomplete CIFAR10 dataset with $n = 1,000$. "RE" denotes the relative error of the estimation, and "Recall" indicates the search accuracy of Recall@10.**

| Metric | RE(S) ↓ | | RE(D) ↓ | | Recall(S) ↑ | | Recall(D) ↑ | |
|--------|---------|------|---------|------|-------------|------|-------------|------|
| Rate $r$ | $S^o$ | EC | $D^o$ | EC | $S^o$ | EC | $D^o$ | EC |
| 20% | 0.041 | **0.039** | 0.013 | **0.012** | 0.927 | **0.931** | 0.919 | **0.925** |
| 30% | 0.056 | **0.051** | 0.017 | **0.016** | 0.903 | **0.912** | 0.893 | **0.904** |
| 40% | 0.073 | **0.064** | 0.023 | **0.020** | 0.876 | **0.892** | 0.863 | **0.883** |
| 50% | 0.095 | **0.078** | 0.029 | **0.025** | 0.842 | **0.871** | 0.823 | **0.859** |
| 60% | 0.126 | **0.096** | 0.038 | **0.030** | 0.795 | **0.846** | 0.769 | **0.830** |
| 70% | 0.175 | **0.120** | 0.052 | **0.038** | 0.724 | **0.811** | 0.687 | **0.793** |
| 80% | 0.269 | **0.156** | 0.079 | **0.049** | 0.605 | **0.758** | 0.543 | **0.734** |

## D.2  Hyperparameter Analysis

For hyperparameter, we select the number of top-$k$ eigenvalues, $k$, from the set $\{1, 5, 10, 15, 20, 25\}$ to enhance search performance, as detailed in Tables D.5 and D.6. Fig. D.10 demonstrates that our method consistently performs well across various $k$ values. It's important to note that the optimal $k$ may vary depending on the specific data types and settings.

**Table D.5: Hyperparameter $k$ of our method on incomplete datasets with $n = 1,000$ and $r = 80\%$.**

| Dataset | CIFAR10 | LFW | COIL100 | ISOLET |
|---------|---------|-----|---------|--------|
| $k$ | 5 | 5 | 1 | 1 |

**Table D.6: Hyperparameter $k$ of our method on incomplete CIFAR10 dataset with $n = 1,000$.**

| Missing Rate $r$ | 20% | 30% | 40% | 50% | 60% | 70% | 80% |
|------------------|-----|-----|-----|-----|-----|-----|-----|
| $k$ | 25 | 20 | 20 | 15 | 10 | 10 | 5 |

## D.3  Efficiency Analysis

• **Efficiency Advantage on Small-Scale Datasets.** Our EC method significantly outpaces imputation techniques in terms of running time. When handling 1,000 incomplete samples, EC is hundreds to thousands of times faster than matrix completion methods like SVT, KFMC, and PMC. It even surpasses the $k$NN method by 2 to 12 times, achieving an impressive runtime of approximately 0.1 seconds across four datasets, as shown in Table D.7.

• **Efficiency Advantage on Large-Scale Datasets.** Our comprehensive testing on the full versions of datasets reveals our method's high efficiency on large-scale datasets. The EC method processes several thousand samples in datasets like LFW, COIL100, and ISOLET in just 3-18 seconds without any partitioning speedup. This performance far exceeds that of traditional imputation methods by several orders of magnitude.

• **Speedup Validation:** Our divide-and-conquer strategy considerably reduces computational complexity to quadratic terms, significantly enhancing efficiency on large datasets, particularly for the CIFAR10 dataset with 30,000 incomplete samples. By implementing a partition size of $m = 1,000$, we achieve a roughly 6-fold

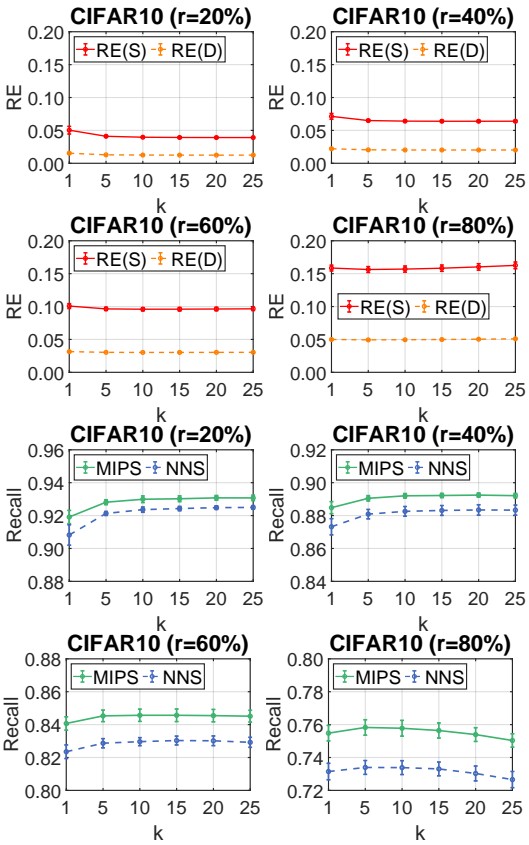

**Figure D.10: Hyperparameter analysis on incomplete CIFAR10 dataset with $n = 1,000$ and various $r$.**

**Table D.7: Efficiency analysis on the incomplete datasets with $n = 1,000$ and $r = 80\%$.**

| Time (sec) | CIFAR10 | LFW | COIL100 | ISOLET |
|------------|---------|-----|---------|--------|
| Mean | $0.05_{\pm0.01}$ | $0.06_{\pm0.00}$ | $0.02_{\pm0.00}$ | $0.01_{\pm0.01}$ |
| $k$NN | $0.68_{\pm0.02}$ | $0.89_{\pm0.02}$ | $0.24_{\pm0.01}$ | $0.16_{\pm0.02}$ |
| SVT | $57.46_{\pm1.65}$ | $77.36_{\pm1.70}$ | $16.97_{\pm0.25}$ | $9.47_{\pm0.10}$ |
| KFMC | $17.23_{\pm19.53}$ | $0.22_{\pm0.02}$ | $14.84_{\pm4.79}$ | $10.97_{\pm1.23}$ |
| PMC | $356.75_{\pm21.24}$ | $378.28_{\pm29.79}$ | $460.81_{\pm9.32}$ | $405.62_{\pm22.42}$ |
| TDM | > 6h | > 6h | > 6h | > 6h |
| GAIN | $792.51_{\pm2.33}$ | $1592.45_{\pm82.16}$ | $102.99_{\pm1.32}$ | $51.39_{\pm3.57}$ |
| MIWAE | $3495.00_{\pm46.87}$ | $4888.27_{\pm235.58}$ | $632.96_{\pm5.26}$ | $398.80_{\pm3.32}$ |
| DMC | $0.06_{\pm0.01}$ | $0.05_{\pm0.00}$ | $0.05_{\pm0.01}$ | $0.05_{\pm0.00}$ |
| SMC | $28.10_{\pm1.35}$ | $26.89_{\pm1.01}$ | $26.56_{\pm0.77}$ | $25.06_{\pm0.09}$ |
| SVC | $0.07_{\pm0.02}$ | $0.06_{\pm0.01}$ | $0.06_{\pm0.00}$ | $0.06_{\pm0.01}$ |
| $S^o, D^o$ | $0.05_{\pm0.00}$ | $0.06_{\pm0.00}$ | $0.02_{\pm0.00}$ | $0.01_{\pm0.00}$ |
| EC (Ours) | $0.08_{\pm0.01}$ | $0.07_{\pm0.01}$ | $0.08_{\pm0.01}$ | $0.07_{\pm0.01}$ |

speedup, completing eigenvalue correction for 30,000 samples in just 4 minutes. This runtime is substantially faster than that of conventional imputation and calibration methods, while the PMC, TDM and SMC methods cannot execute within 6 hours.

**Table D.8: Performance of Maximum Inner Product Search on Four Entire Datasets with $r = 80\%$. "RE(S)" denotes the relative error of the inner product estimation, and "Recall(S)" indicates the MIPS accuracy of Recall 10@10. Bold highlights the best result. The last line uses "↓" to indicate error reduction from $S^o$ to ours, and "↑" to show accuracy improvement from $S^o$ to ours.**

| Dataset | CIFAR10 | | LFW | | COIL100 | | ISOLET | |
| Size $n$ | 30,000 | | 6,600 | | 3,600 | | 3,800 | |
| Metric | RE(S) ↓ | Recall(S) ↑ | RE(S) ↓ | Recall(S) ↑ | RE(S) ↓ | Recall(S) ↑ | RE(S) ↓ | Recall(S) ↑ |
|---|---|---|---|---|---|---|---|---|
| Mean | $0.960_{\pm0.000}$ | $0.307_{\pm0.003}$ | $0.960_{\pm0.000}$ | $0.409_{\pm0.004}$ | $0.959_{\pm0.000}$ | $0.231_{\pm0.003}$ | $0.960_{\pm0.000}$ | $0.139_{\pm0.002}$ |
| $k$NN | $0.942_{\pm0.001}$ | $0.328_{\pm0.007}$ | $0.948_{\pm0.001}$ | $0.443_{\pm0.005}$ | $0.941_{\pm0.002}$ | $0.249_{\pm0.005}$ | $0.943_{\pm0.001}$ | $0.157_{\pm0.002}$ |
| SVT | $0.866_{\pm0.000}$ | $0.235_{\pm0.008}$ | $0.867_{\pm0.000}$ | $0.288_{\pm0.005}$ | $0.870_{\pm0.001}$ | $0.219_{\pm0.005}$ | $0.882_{\pm0.000}$ | $0.125_{\pm0.003}$ |
| KFMC | $0.929_{\pm0.001}$ | $0.119_{\pm0.034}$ | $0.943_{\pm0.009}$ | $0.460_{\pm0.031}$ | $0.919_{\pm0.002}$ | $0.107_{\pm0.008}$ | $0.933_{\pm0.001}$ | $0.145_{\pm0.007}$ |
| $S^o$ | $0.270_{\pm0.001}$ | $0.362_{\pm0.002}$ | $0.273_{\pm0.002}$ | $0.451_{\pm0.004}$ | $0.579_{\pm0.004}$ | $0.240_{\pm0.004}$ | $0.816_{\pm0.007}$ | $0.135_{\pm0.002}$ |
| DMC | $0.253_{\pm0.001}$ | $0.395_{\pm0.002}$ | $0.239_{\pm0.001}$ | $0.528_{\pm0.003}$ | $0.517_{\pm0.004}$ | $0.277_{\pm0.005}$ | $0.750_{\pm0.006}$ | $0.160_{\pm0.002}$ |
| SMC | - | - | - | - | $0.367_{\pm0.002}$ | $0.296_{\pm0.005}$ | $0.508_{\pm0.004}$ | $0.178_{\pm0.003}$ |
| SVC | $0.231_{\pm0.008}$ | $0.416_{\pm0.018}$ | $0.218_{\pm0.004}$ | $0.536_{\pm0.009}$ | $0.473_{\pm0.014}$ | $0.284_{\pm0.004}$ | $0.689_{\pm0.020}$ | $0.167_{\pm0.002}$ |
| EC (Ours) | $\mathbf{0.141_{\pm0.000}}$ | $\mathbf{0.617_{\pm0.003}}$ | $\mathbf{0.146_{\pm0.001}}$ | $\mathbf{0.673_{\pm0.003}}$ | $\mathbf{0.290_{\pm0.002}}$ | $\mathbf{0.360_{\pm0.007}}$ | $\mathbf{0.379_{\pm0.004}}$ | $\mathbf{0.277_{\pm0.004}}$ |
| $EC$-5,000 | $0.144_{\pm0.000}$ | $0.602_{\pm0.002}$ | - | - | - | - | - | - |
| $EC$-2,000 | $0.149_{\pm0.000}$ | $0.585_{\pm0.002}$ | - | - | - | - | - | - |
| $EC$-1,000 | $0.155_{\pm0.000}$ | $0.564_{\pm0.002}$ | - | - | - | - | - | - |
| $S^o \rightarrow$ EC | 48%↓$_{\pm0\%}$ | 70%↑$_{\pm1\%}$ | 47%↓$_{\pm0\%}$ | 49%↑$_{\pm1\%}$ | 50%↓$_{\pm1\%}$ | 50%↑$_{\pm3\%}$ | 53%↓$_{\pm0\%}$ | 106%↑$_{\pm4\%}$ |

**Table D.9: Performance of Nearest Neighbor Search on Four Entire Datasets with $r = 80\%$. "RE(D)" denotes the relative error of the Euclidean distance estimation, and "Recall(D)" indicates the NNS accuracy of Recall 10@10. Bold highlights the best result. The last line uses "↓" to indicate error reduction from $D^o$ to ours, and "↑" to show accuracy improvement from $D^o$ to ours.**

| Dataset | CIFAR10 | | LFW | | COIL100 | | ISOLET | |
| Size $n$ | 30,000 | | 6,600 | | 3,600 | | 3,800 | |
| Metric | RE(D) ↓ | Recall(D) ↑ | RE(D) ↓ | Recall(D) ↑ | RE(D) ↓ | Recall(D) ↑ | RE(D) ↓ | Recall(D) ↑ |
|---|---|---|---|---|---|---|---|---|
| Mean | $0.814_{\pm0.000}$ | $0.032_{\pm0.002}$ | $0.811_{\pm0.000}$ | $0.079_{\pm0.004}$ | $0.809_{\pm0.002}$ | $0.026_{\pm0.002}$ | $0.810_{\pm0.001}$ | $0.037_{\pm0.003}$ |
| $k$NN | $0.809_{\pm0.000}$ | $0.033_{\pm0.002}$ | $0.807_{\pm0.000}$ | $0.081_{\pm0.005}$ | $0.802_{\pm0.002}$ | $0.033_{\pm0.002}$ | $0.802_{\pm0.001}$ | $0.044_{\pm0.004}$ |
| SVT | $0.790_{\pm0.000}$ | $0.061_{\pm0.003}$ | $0.791_{\pm0.000}$ | $0.137_{\pm0.004}$ | $0.790_{\pm0.002}$ | $0.068_{\pm0.002}$ | $0.795_{\pm0.001}$ | $0.068_{\pm0.004}$ |
| KFMC | $0.804_{\pm0.001}$ | $0.047_{\pm0.003}$ | $0.808_{\pm0.001}$ | $0.100_{\pm0.013}$ | $0.775_{\pm0.005}$ | $0.057_{\pm0.002}$ | $0.802_{\pm0.002}$ | $0.069_{\pm0.003}$ |
| $D^o$ | $0.079_{\pm0.000}$ | $0.245_{\pm0.001}$ | $0.072_{\pm0.000}$ | $0.389_{\pm0.002}$ | $0.216_{\pm0.012}$ | $0.259_{\pm0.002}$ | $0.225_{\pm0.007}$ | $0.121_{\pm0.002}$ |
| DMC | $3.048_{\pm0.002}$ | $0.035_{\pm0.002}$ | $1.808_{\pm0.002}$ | $0.159_{\pm0.007}$ | $2.378_{\pm0.036}$ | $0.049_{\pm0.002}$ | $3.112_{\pm0.017}$ | $0.063_{\pm0.003}$ |
| SMC | - | - | - | - | $1.102_{\pm0.017}$ | $0.255_{\pm0.003}$ | $1.566_{\pm0.012}$ | $0.180_{\pm0.003}$ |
| SVC | $2.667_{\pm0.002}$ | $0.034_{\pm0.002}$ | $1.281_{\pm0.002}$ | $0.223_{\pm0.007}$ | $1.758_{\pm0.027}$ | $0.086_{\pm0.006}$ | $2.538_{\pm0.013}$ | $0.085_{\pm0.004}$ |
| EC (Ours) | $\mathbf{0.046_{\pm0.000}}$ | $\mathbf{0.567_{\pm0.001}}$ | $\mathbf{0.043_{\pm0.000}}$ | $\mathbf{0.650_{\pm0.003}}$ | $\mathbf{0.183_{\pm0.014}}$ | $\mathbf{0.356_{\pm0.011}}$ | $\mathbf{0.154_{\pm0.009}}$ | $\mathbf{0.276_{\pm0.003}}$ |
| $EC$-5,000 | $0.046_{\pm0.000}$ | $0.552_{\pm0.001}$ | - | - | - | - | - | - |
| $EC$-2,000 | $0.048_{\pm0.000}$ | $0.534_{\pm0.001}$ | - | - | - | - | - | - |
| $EC$-1,000 | $0.049_{\pm0.000}$ | $0.511_{\pm0.002}$ | - | - | - | - | - | - |
| $D^o \rightarrow$ EC | 42%↓$_{\pm0\%}$ | 131%↑$_{\pm1\%}$ | 40%↓$_{\pm0\%}$ | 67%↑$_{\pm1\%}$ | 15%↓$_{\pm2\%}$ | 37%↑$_{\pm4\%}$ | 32%↓$_{\pm2\%}$ | 127%↑$_{\pm4\%}$ |

## D.4 Scalability Analysis

• **Scalability Evidence.** Our method significantly enhances scalability and performance on large datasets. For example, Tables D.8 and D.9 illustrate the EC method's efficacy on the large-scale CIFAR10 dataset with 30,000 samples. Both with and without partitioning, the EC method exhibits exceptional scalability, achieving the smallest estimation errors and the highest recall values.

• **Performance Advantage.** On the full-version large datasets, the similarity search performance of some imputation methods drops markedly, with Recall(D) falling below 0.1. In contrast, our EC method maintains robust performance; even the worst results, with a partition size of $m = 1,000$, still achieve Recall above 0.5

on the CIFAR10 dataset. This significantly surpasses all baseline methods, underscoring our method's effectiveness and scalability.

## D.5 Extension on Various Missing Mechanisms

To align with baseline methods, we adopt the MCAR mechanism for main experiments. Furthermore, we show the effectiveness of our method on other missing mechanisms, as discussed in Section 6.5. Specifically, we incorporate realistic missing data patterns [5] in the CIFAR10 dataset, detailed below:

---

[5] Official codes [28] from https://github.com/hezgit/TDM were used to simulate the MAR and MNAR.

**Table D.10: Performance of Maximum Inner Product Search under Various Missing Mechanisms on incomplete CIFAR10 dataset with $n = 1,000$ and $r = 80\%$. "RE(S)" denotes the relative error of the inner product estimation, and "Recall(S)" indicates the MIPS accuracy of Recall 10@10. Bold highlights the best result. The last line uses "↓" to indicate error reduction from $S^o$ to ours, and "↑" to show accuracy improvement from $S^o$ to ours.**

| Mechanism | MAR | | MNAR | | Segmental-Missing | | Block-Missing | |
|---|---|---|---|---|---|---|---|---|
| Metric | RE(S) ↓ | Recall(S) ↑ | RE(S) ↓ | Recall(S) ↑ | RE(S) ↓ | Recall(S) ↑ | RE(S) ↓ | Recall(S) ↑ |
| Mean | $0.886_{\pm0.003}$ | $0.786_{\pm0.012}$ | $0.955_{\pm0.001}$ | $0.656_{\pm0.008}$ | $0.958_{\pm0.000}$ | $0.527_{\pm0.014}$ | $0.953_{\pm0.000}$ | $0.390_{\pm0.012}$ |
| $k$NN | $0.875_{\pm0.003}$ | $0.790_{\pm0.012}$ | $0.942_{\pm0.002}$ | $0.675_{\pm0.009}$ | $0.946_{\pm0.003}$ | $0.570_{\pm0.010}$ | $0.939_{\pm0.003}$ | $0.440_{\pm0.014}$ |
| SVT | $0.802_{\pm0.005}$ | $0.673_{\pm0.012}$ | $0.854_{\pm0.004}$ | $0.518_{\pm0.007}$ | $0.866_{\pm0.003}$ | $0.476_{\pm0.012}$ | $0.856_{\pm0.004}$ | $0.433_{\pm0.013}$ |
| KFMC | $0.845_{\pm0.028}$ | $0.664_{\pm0.092}$ | $0.922_{\pm0.028}$ | $0.604_{\pm0.056}$ | $0.948_{\pm0.014}$ | $0.560_{\pm0.038}$ | $0.922_{\pm0.034}$ | $0.358_{\pm0.039}$ |
| PMC | $0.755_{\pm0.012}$ | $0.604_{\pm0.019}$ | $0.801_{\pm0.012}$ | $0.510_{\pm0.020}$ | $0.829_{\pm0.011}$ | $0.426_{\pm0.022}$ | $0.718_{\pm0.019}$ | $0.503_{\pm0.021}$ |
| TDM | $0.885_{\pm0.003}$ | $0.646_{\pm0.015}$ | $0.954_{\pm0.002}$ | $0.415_{\pm0.024}$ | $0.957_{\pm0.001}$ | $0.276_{\pm0.027}$ | $0.952_{\pm0.000}$ | $0.270_{\pm0.022}$ |
| $S^o$ | $0.158_{\pm0.014}$ | $0.792_{\pm0.013}$ | $0.264_{\pm0.007}$ | $0.640_{\pm0.005}$ | $0.310_{\pm0.008}$ | $0.561_{\pm0.011}$ | $0.518_{\pm0.013}$ | $0.424_{\pm0.011}$ |
| DMC | $0.154_{\pm0.015}$ | $0.798_{\pm0.014}$ | $0.221_{\pm0.005}$ | $0.705_{\pm0.003}$ | $0.265_{\pm0.007}$ | $0.630_{\pm0.011}$ | $0.465_{\pm0.012}$ | $0.479_{\pm0.011}$ |
| SMC | $0.153_{\pm0.015}$ | $0.801_{\pm0.014}$ | $0.182_{\pm0.004}$ | $0.736_{\pm0.004}$ | $0.219_{\pm0.006}$ | $0.670_{\pm0.010}$ | $0.392_{\pm0.012}$ | $0.524_{\pm0.011}$ |
| SVC | $0.195_{\pm0.050}$ | $0.695_{\pm0.078}$ | $0.223_{\pm0.010}$ | $0.650_{\pm0.043}$ | $0.255_{\pm0.006}$ | $0.597_{\pm0.026}$ | $0.437_{\pm0.014}$ | $0.462_{\pm0.016}$ |
| EC (Ours) | $\mathbf{0.152}_{\pm0.015}$ | $\mathbf{0.802}_{\pm0.014}$ | $\mathbf{0.155}_{\pm0.003}$ | $\mathbf{0.770}_{\pm0.004}$ | $\mathbf{0.190}_{\pm0.004}$ | $\mathbf{0.711}_{\pm0.009}$ | $\mathbf{0.342}_{\pm0.020}$ | $\mathbf{0.565}_{\pm0.012}$ |
| $S^o \rightarrow$ EC | 4%↓$_{\pm1\%}$ | 1%↑$_{\pm0\%}$ | 41%↓$_{\pm1\%}$ | 20%↑$_{\pm1\%}$ | 39%↓$_{\pm1\%}$ | 27%↑$_{\pm1\%}$ | 34%↓$_{\pm3\%}$ | 33%↑$_{\pm2\%}$ |

**Table D.11: Performance of Nearest Neighbor Search under Various Missing Mechanisms on incomplete CIFAR10 dataset with $n = 1,000$ and $r = 80\%$. "RE(D)" denotes the relative error of the Euclidean distance estimation, and "Recall(D)" indicates the NNS accuracy of Recall 10@10. Bold highlights the best result. The last line uses "↓" to indicate error reduction from $D^o$ to ours, and "↑" to show accuracy improvement from $D^o$ to ours.**

| Mechanism | MAR | | MNAR | | Segmental-Missing | | Block-Missing | |
|---|---|---|---|---|---|---|---|---|
| Metric | RE(D) ↓ | Recall(D) ↑ | RE(D) ↓ | Recall(D) ↑ | RE(D) ↓ | Recall(D) ↑ | RE(D) ↓ | Recall(D) ↑ |
| Mean | $0.799_{\pm0.001}$ | $0.481_{\pm0.010}$ | $0.808_{\pm0.001}$ | $0.187_{\pm0.012}$ | $0.814_{\pm0.001}$ | $0.192_{\pm0.013}$ | $0.813_{\pm0.001}$ | $0.190_{\pm0.009}$ |
| $k$NN | $0.795_{\pm0.000}$ | $0.501_{\pm0.010}$ | $0.803_{\pm0.001}$ | $0.194_{\pm0.013}$ | $0.810_{\pm0.001}$ | $0.197_{\pm0.014}$ | $0.809_{\pm0.001}$ | $0.198_{\pm0.009}$ |
| SVT | $0.772_{\pm0.002}$ | $0.581_{\pm0.008}$ | $0.780_{\pm0.001}$ | $0.330_{\pm0.013}$ | $0.790_{\pm0.001}$ | $0.303_{\pm0.010}$ | $0.788_{\pm0.001}$ | $0.300_{\pm0.012}$ |
| KFMC | $0.770_{\pm0.022}$ | $0.524_{\pm0.034}$ | $0.788_{\pm0.017}$ | $0.220_{\pm0.032}$ | $0.811_{\pm0.004}$ | $0.211_{\pm0.026}$ | $0.797_{\pm0.018}$ | $0.222_{\pm0.035}$ |
| PMC | $0.720_{\pm0.010}$ | $0.607_{\pm0.010}$ | $0.713_{\pm0.010}$ | $0.322_{\pm0.012}$ | $0.737_{\pm0.008}$ | $0.310_{\pm0.016}$ | $0.686_{\pm0.013}$ | $0.378_{\pm0.014}$ |
| TDM | $0.777_{\pm0.001}$ | $0.377_{\pm0.017}$ | $0.784_{\pm0.002}$ | $0.158_{\pm0.011}$ | $0.790_{\pm0.004}$ | $0.160_{\pm0.014}$ | $0.790_{\pm0.002}$ | $0.163_{\pm0.015}$ |
| $D^o$ | $0.050_{\pm0.004}$ | $0.753_{\pm0.013}$ | $0.078_{\pm0.001}$ | $0.562_{\pm0.007}$ | $0.092_{\pm0.001}$ | $0.499_{\pm0.006}$ | $0.154_{\pm0.001}$ | $0.333_{\pm0.006}$ |
| DMC | $0.139_{\pm0.003}$ | $0.743_{\pm0.012}$ | $0.719_{\pm0.006}$ | $0.462_{\pm0.012}$ | $0.813_{\pm0.006}$ | $0.419_{\pm0.014}$ | $1.186_{\pm0.008}$ | $0.298_{\pm0.011}$ |
| SMC | $0.068_{\pm0.004}$ | $0.722_{\pm0.012}$ | $0.296_{\pm0.006}$ | $0.473_{\pm0.028}$ | $0.341_{\pm0.008}$ | $0.438_{\pm0.023}$ | $0.522_{\pm0.013}$ | $0.359_{\pm0.020}$ |
| SVC | $0.096_{\pm0.009}$ | $0.722_{\pm0.040}$ | $0.476_{\pm0.003}$ | $0.523_{\pm0.013}$ | $0.544_{\pm0.004}$ | $0.477_{\pm0.013}$ | $0.829_{\pm0.005}$ | $0.345_{\pm0.010}$ |
| EC (Ours) | $\mathbf{0.048}_{\pm0.004}$ | $\mathbf{0.763}_{\pm0.014}$ | $\mathbf{0.049}_{\pm0.001}$ | $\mathbf{0.734}_{\pm0.007}$ | $\mathbf{0.061}_{\pm0.001}$ | $\mathbf{0.689}_{\pm0.006}$ | $\mathbf{0.108}_{\pm0.004}$ | $\mathbf{0.529}_{\pm0.011}$ |
| $D^o \rightarrow$ EC | 3%↓$_{\pm0\%}$ | 1%↑$_{\pm0\%}$ | 37%↓$_{\pm1\%}$ | 31%↑$_{\pm1\%}$ | 34%↓$_{\pm1\%}$ | 38%↑$_{\pm1\%}$ | 30%↓$_{\pm2\%}$ | 59%↑$_{\pm4\%}$ |

• **Missing at Random (MAR) [21]:** It samples a fixed subset of features to remain complete, while the rest are subject to missingness based on a logistic model using the non-missing features as inputs.

• **Missing Not at Random (MNAR) [28]:** It implements a logistic model where inputs are masked by MCAR, creating a logistic-masking MNAR pattern.

• **Segmental-Missing (SM):** Pixels in vectorized images are missing in segments of random lengths.

• **Block-Missing (BM):** Pixels in original $32 \times 32$ images are missing in blocks of random sizes.

The results in Tables D.10 and D.11 highlight our method's robustness and effectiveness across a range of realistic missing data scenarios, characterized by the smallest estimation errors and highest search accuracy. While the initial estimate $S^o$ ($D^o$) shows good performance under the MAR setting, leaving little room for improvement, our method significantly outperforms the initial estimate and baseline methods in MNAR, Segmental-Missing, and Block-Missing.

