# OpenReview forum: "A Theory-Driven Approach to Inner Product Matrix Estimation for Incomplete Data: An Eigenvalue Perspective"
_ACM.org/TheWebConf/2025/Conference — WWW 2025 Poster_

### Official Review · Reviewer_MVgR · 2024-11-22

**Novelty:** 5
**Technical Quality:** 6

**Review:**

**Summary**
- This paper addresses the task of estimating inner product matrices from incomplete data, which is crucial for applications like maximum inner product search and nearest neighbor search. The authors propose a novel eigenvalue correction method based on random matrix theory, specifically leveraging the Marchenko-Pastur Law, to adjust the eigenvalue distribution of the estimated inner product matrix derived from incomplete data.

**Strengths**
- The paper clearly targets challenges in existing methods by introducing a theory-driven approach, which is novel in its focus on correcting the eigenvalue distortions in inner product matrices caused by missing data.
- The proposed approach is simple yet effective, and the paper provides rigorous theoretical analysis including optimality proofs and error bounds, which strengthen the validity of the proposed method.
- Extensive experiments on multiple datasets demonstrate the significant improvements over baselines in both estimation error and similarity search tasks, especially under high missing rate scenarios.
- The method could be extended to non-i.i.d. and real-world data without relying on specific missing mechanisms, demonstrating its potential applicability across various scenarios.

**Weaknesses**
- The robustness of the method across diverse missing mechanisms, such as MNAR, could be better emphasized by detailing how its design or strengths enable this generalization beyond the MCAR assumption. The theoretical development of the method is based on the assumption of the MCAR scenario, which makes it somewhat unclear how it can also handle more complex scenarios like MNAR. Although the authors’ claim of applicability to various missing mechanisms is supported by experimental results and appears reasonable, a more detailed explanation of which strengths or aspects of their method enable this generalization would help clarify the strength of proposed method.
- Some of the method descriptions and notations are complex and may be challenging to follow for readers not deeply familiar with random matrix theory. For example, the application of the Marchenko-Pastur Law in Section 3.1 may be challenging for readers unfamiliar with the task. While its role in determining spectral bounds is crucial, the explanation is heavily mathematical and lacks intuitive clarification. A brief overview of the law’s significance in eigenvalue correction would make this section more accessible and improve readability.
- While the paper provides a strong theoretical foundation, it could benefit from a deeper discussion of potential failure cases or scenarios not fully addressed by the current approach. For instance, including a discussion on the impact of neglecting eigenvector correction (as the method focuses solely on eigenvalue correction) and its potential consequences in tasks such as clustering or dimensionality reduction would further strengthen the paper’s contribution.

**Questions:**

1. In the experiments, SMC has several variants that were not included as baselines. Were these variants excluded because they are not applicable in the experiments conducted on this paper?
2. While the paper focuses on correcting eigenvalues and mentions in Section 1 that accurate estimation of high-dimensional eigenvectors is challenging, could the authors elaborate on whether they considered addressing eigenvector correction as well? Even if challenging, might correcting eigenvectors further improve the reconstruction of the true inner product matrix, and is this a problem worth exploring in future research?

**Reviewer Confidence:**

2: The reviewer is willing to defend the evaluation, but it is likely that the reviewer did not understand parts of the paper

**Scope:**

3: The work is somewhat relevant to the Web and to the track, and is of narrow interest to a sub-community

---

### Official Review · Reviewer_4bGZ · 2024-11-28

**Novelty:** 4
**Technical Quality:** 4

**Review:**

To address the challenge of estimating inner product matrices from incomplete data, this paper focuses on eigenvalue corrections. By leveraging random matrix theory, specifically the Marchenko-Pastur Law, this paper proposes an eigenvalue correction method to adjust the eigenvalue distribution of the estimated matrix. This method aims to enhance the accuracy of both inner product and Euclidean distance matrices, which are crucial for tasks like maximum inner product search (MIPS) and nearest neighbor search (NNS), particularly in scenarios with high missingness. Specifically, the paper offers a theoretically grounded approach using random matrix theory to correct the eigenvalue distribution. The method shows robust performance in data imputation and similarity calibration, especially in handling high missing rates. The method is applied to both i.i.d. and non-i.i.d. data.

**Questions:**

Questions about non-i.i.d. data

(Q1.1) For non-i.i.d. data discussed in Sec. 4, what is the theoretical guarantee for the eigenvalue correction strategy when applied to non-i.i.d. data?

(Q1.2) Is the proposed method sensitive to the choice of the k used in the correction process? For the k in recall@k in experiments are all set as k=10.

(Q1.3) Since the authors discuss inner product estimation for i.i.d. data and non-i.i.d. data separately in Secs. 3 and 4, the authors should explain whether the method can be used without assuming a specific data distribution, that is, when we do not know whether the data is i.i.d. or non-i.i.d.

(Q2) Please provide code of the proposed methods.

**Reviewer Confidence:**

3: The reviewer is confident but not certain that the evaluation is correct

**Scope:**

4: The work is relevant to the Web and to the track, and is of broad interest to the community

---

### Official Review · Reviewer_sjyn · 2024-11-29

**Novelty:** 5
**Technical Quality:** 5

**Review:**

This paper completes incomplete data by analyzing the eigenvalue distributions and uses inner product and Euclidean distance as a measure of completion effectiveness. Since inner product calculations directly impact certain search tasks like MIPS and NNS, the proposed method in this paper provides a more intuitive and effective way for search tasks of incomplete data.

Pros:

1.	This paper offers highly detailed mathematical proofs that are well-theorized.

2.	The final experimental evaluation is conducted from multiple perspectives, and rich experiments are also performed on intermediate results to validate the theoretical findings.

Cons:

1.	Some of the math proofs are complex and might be hard for readers without strong background of random matrix theory. Adding simpler explanations or examples could make it easier to understand.

**Questions:**

1.	Figure 4 is not referenced anywhere in the entire paper.

2.	Since I am not familiar with this field, it is difficult to understand the relationship between separable data and non-i.i.d. data solely based on the content of the paper. Also, it would be beneficial to include some derivation insights, such as how to transform from Lemma 1 to Theorem 2. This will make the understanding more coherent and seamless for the readers.

**Reviewer Confidence:**

2: The reviewer is willing to defend the evaluation, but it is likely that the reviewer did not understand parts of the paper

**Scope:**

3: The work is somewhat relevant to the Web and to the track, and is of narrow interest to a sub-community

---

### Official Review · Reviewer_WKGS · 2024-12-02

**Novelty:** 6
**Technical Quality:** 6

**Review:**

The authors of this paper observe that the eigenvalue distribution of datasets follows that of the complete dataset, even when different percentages of missing data are present. Based on this observation, they propose a new method for filling in missing data. The experimental results demonstrate that their approach significantly outperforms various baselines. They also show that their method does not depend on a specific mechanism used to generate missing data in the testing datasets.

Pros:
1. This paper presents an interesting method for addressing an important research topic in ML applications: filling in missing data.
2. The authors provide detailed theoretical and empirical evidence supporting the effectiveness of their framework.
3. The experimental results show significant improvements over baselines.

Cons:
1. The paper is somewhat difficult to read for audiences without a strong linear algebra background, as it lacks explanations for some key concepts.
2. The paper focuses on demonstrating their framework from a mathematical perspective, with little effort to connect these mathematical concepts to the research problem at hand.
3. The authors begin with i.i.d. datasets, then move on to separable datasets, and finally extend their work to general datasets, but they provide limited introduction to the overall argument structure. This makes it easy for readers to feel lost during the progression of the argument.

**Questions:**

1. The authors did not adequately explain concepts such as "low-rank," "high-rank," "optimal transport-based methods," and "PSD." This makes it difficult for readers without a related background to follow the argument. For example, why should PSD be emphasized? What are the authors trying to convey by presenting "Non-PSD" and "PSD" in Figure 1?
2. The authors heavily reference random matrix theory without providing detailed explanations. I suggest adding necessary citations and a brief explanation of this theory.
3. Are there better ways to introduce concepts like "ESD" and "LSD" for improved accessibility?
4. In Section 3.2, are there clearer ways to connect the observations of the "Impact of r" and "Impact of c" to the research problem? What is the implication of the distribution being more sensitive to smaller **c**?
5. It remains unclear how refill data is generated from the estimated eigenvalue distribution until Section 4.3. I suggest clarifying the overall argument structure earlier in the paper.
6. Theorem 6 is not proven in the Appendix, and it is not immediately clear why it follows from the MP law.
7. Can the authors provide a clearer definition of a separable dataset in Section 4, Paragraph 2?
8. The authors tested their method on image and speech datasets; did they evaluate it on text datasets?
9. It would be helpful to add more explanation about the baselines. Are any of these baselines based on association rules, or do they attempt to fill missing values by drawing from an underlying distribution? Can the authors compare their method to these baselines as well?
10. In Table 2, why did the authors choose Recall@10 rather than Precision@10 to evaluate these methods?

**Reviewer Confidence:**

2: The reviewer is willing to defend the evaluation, but it is likely that the reviewer did not understand parts of the paper

**Scope:**

4: The work is relevant to the Web and to the track, and is of broad interest to the community

---

### Official Review · Reviewer_Zt79 · 2024-12-02

**Novelty:** 5
**Technical Quality:** 6

**Review:**

This paper introduces a novel approach for estimating inner product matrices from incomplete data by aligning the eigenvalue distribution of the estimated matrix with the true distribution using insights from random matrix theory.

### Strengths

- Theory-driven approach based on random matrix theory
- Simple yet effective algorithms for eigenvalue correction
- Comprehensive empirical validation on various datasets and missingness scenarios
- Improved performance on similarity search tasks

### Weaknesses

- The experimental evaluation is limited to image datasets (CIFAR10, LFW, COIL100) and does not include any standard IR test collections like TREC, MSMARCO, MovieLens etc. This raises questions about the generalizability of the results to real-world IR applications and makes it harder to compare to other IR methods.
- The empirical assessment focuses narrowly on the MIPS and NNS similarity search setups. A more comprehensive evaluation on downstream IR tasks such as ranking or recommendation would provide stronger evidence of the methods' practical impact.

**Questions:**

1. The experiments are conducted exclusively on image datasets. Could you discuss how the characteristics of these datasets might differ from typical text-based data encountered in IR?
2. The evaluation focuses on the MIPS and NNS search problems. While these are important components of many IR pipelines, could you comment on how the improved inner product and distance estimation might directly benefit specific downstream IR applications like document ranking, collaborative filtering, or clustering?

**Reviewer Confidence:**

1: The reviewer's evaluation is an educated guess

**Scope:**

3: The work is somewhat relevant to the Web and to the track, and is of narrow interest to a sub-community